# Cell type-specific modulation of sensory and affective components of itch in the periaqueductal gray

Vijay K. Samineni [1,2], Jose G. Grajales-Reyes [1,2,3,4], Saranya S. Sundaram[1,2], Judy J. Yoo [1,2] & Robert W. Gereau IV [1,2,5]

Itch is a distinct aversive sensation that elicits a strong urge to scratch. Despite recent advances in our understanding of the peripheral basis of itch, we know very little regarding how central neural circuits modulate acute and chronic itch processing. Here we establish the causal contributions of defined periaqueductal gray (PAG) neuronal populations in itch modulation in mice. Chemogenetic manipulations demonstrate bidirectional modulation of scratching by neurons in the PAG. Fiber photometry studies show that activity of GABAergic and glutamatergic neurons in the PAG is modulated in an opposing manner during chloroquine-evoked scratching. Furthermore, activation of PAG GABAergic neurons or inhibition of glutamatergic neurons resulted in attenuation of scratching in both acute and chronic pruritis. Surprisingly, PAG GABAergic neurons, but not glutamatergic neurons, may encode the aversive component of itch. Thus, the PAG represents a neuromodulatory hub that regulates both the sensory and affective aspects of acute and chronic itch.

[1] Department of Anesthesiology, Washington University School of Medicine, 660 S. Euclid Ave, Box 8054, St. Louis, MO 63110, USA. [2] Washington University Pain Center, Washington University School of Medicine, 660 S. Euclid Ave, Box 8054, St. Louis, MO 63110, USA. [3] Medical Scientist Training Program, Washington University School of Medicine, 660 S. Euclid Ave, Box 8054, St. Louis, MO 63110, USA. [4] Neuroscience Program, Washington University School of Medicine, 660 S. Euclid Ave, Box 8054, St. Louis, MO 63110, USA. [5] Department of Neuroscience, Department of Biomedical Engineering, Washington University School of Medicine, 660 S. Euclid Ave, Box 8054, St. Louis, MO 63110, USA. Correspondence and requests for materials should be addressed to R.W.G.IV. (email: gereaur@wustl.edu)

I tch is a distinct unpleasant sensation that evokes a strong desire to scratch[1]. While acute itch is posited to play an important physiological role as a defense mechanism, chronic itch disorders are disabling and often difficult to treat. Chronic itch (pruritus) disorders affect 15% of the population are often associated with neuropathy, skin disease, and organ dysfunction and are the most frequent cause of visits to dermatologists[2]. Antihistamines are the first-line treatment for itch, but fail to control most forms of chronic pruritus[2]. Recently, non-histaminergic pathways for itch have been identified. Effective alternative therapies that target non-histaminergic pathways are not yet available[3–5].

The neural pathways that signal itch and pain appear remarkably similar, and significant advances have recently been made to tease apart itch and pain circuitry at the level of the primary afferent fibers and the spinal cord[3–5]. However, whether itch and pain information are integrated into the same or different neuronal populations is currently a matter of controversy. In agreement with the concept of shared processing nodes, recent studies support the notion of a robust itch modulatory system within regions of the nervous system that are related to nociception[6–13]. For example, scratching the skin reduces the firing rate of primate spinothalamic tract (STT) neurons responding to pruritogens, but does not alter neuronal responses to painful stimuli[14]. In addition, inhibition or transection of the upper cervical spinal cord reduces scratch-evoked inhibition of spinal cord neuronal activity induced by pruritic stimuli, suggesting that the inhibitory effect of scratching is due to the engagement of a strong descending supraspinal influence on spinal itch processing[15]. The nature of this supraspinal itch modulatory system is not known, but the similarities in anatomical pathways for pain and itch provide a framework for testable hypotheses.

Neuroimaging studies suggest that the periaqueductal gray, an evolutionarily conserved structure in the midbrain, may mediate the suppression of itch induced by noxious counter-stimuli such as cold and scratching[16–18]. The ventrolateral periaqueductal gray (vlPAG) has been shown to regulate several complex behaviors including pain[19] and is a major site of endogenous opioid-induced pain suppression[20]. Electrical stimulation of the PAG produces profound analgesia[21,22] by modulating spinal nociceptive transmission in part by descending projections to the rostral ventromedial medulla (RVM), which sequentially projects to the spinal cord[23–26]. It has been demonstrated in cats that[27] the magnitude of the response to a pruritic stimulus is greatly diminished following decerebration, suggesting the existence of a descending itch modulation system. Consistent with this idea, electrical stimulation of the PAG results in inhibition of spinal neurons responsive to intracutaneous delivery of histamine[28], and this inhibition is lost upon termination of PAG stimulation,. Although the importance of the PAG in pruritic modulation is established, very little is known regarding how the PAG participates in pruritic processing. An understanding of the midbrain neural circuits and transmitters that underlie the suppression of itch may identify novel targets for the treatment of pruritus, in a manner similar to the advancement of opioid receptor pharmacology from our understanding of the role of the PAG in pain modulation[29–32].

Here, we use pharmacological, cell type-specific chemogenetic manipulations and optical imaging approaches to functionally identify and dissect the involvement of specific PAG neuronal subpopulations in the modulation of acute and chronic itch. The results that follow highlight the bidirectional modulation of scratching behaviors by PAG neurons, with opposing roles of glutamatergic and GABAergic subpopulations in mediating the sensory component of itch. Further, we identify a unique role for GABAergic PAG neurons in the encoding or modulation of itch-related aversion.

## Results

### Bidirectional modulation of itch-related behaviors by vlPAG.
To determine the overall contribution of the vlPAG to itch processing, we performed bilateral microinjection of 4% lidocaine (a short acting Na[+] channel blocker) into the vlPAG to block neuronal activity during pruritogen-induced scratching. Bilateral lidocaine microinjection into the vlPAG significantly attenuated scratching evoked by intradermal injection of the non-histaminergic pruritogen chloroquine (200 µg/50 µl) (Fig. 1a, e, f). Vehicle microinjection into the vlPAG did not produce any changes in chloroquine evoked scratching (Fig. 1d, f).

While this result suggests the importance of the vlPAG in regulating pruritogen-evoked scratching, the nature of lidocaine action raises the question of whether resident neurons within the vlPAG or fibers of passage within the vlPAG contribute to itch processing. We therefore turned to a chemogenetic approach to selectively activate or inhibit resident vlPAG neurons. Adeno-Associated Virus type 8 (AAV8) vectors carrying neuron-specific stimulatory (hM3Dq) or inhibitory (hM4Di) DREADDs fused with mCherry (AAV8-hSyn-hM3Dq/hM4Di-mCherry)[33] were microinjected bilaterally into the vlPAG (Fig. 2a, e). In animals expressing the stimulatory DREAAD (hM3Dq), clozapine-N-oxide (CNO) injection (1 mg/kg, i.p) resulted in a significant reduction in chloroquine evoked scratching (Fig. 2b, c,). This reduction was dose dependent (Supplementary Fig. 1a). As a control, and to rule out any possibility of CNO having any effect on chloroquine evoked scratching, we administered CNO to mice injected with an EGFP control virus (AAV8-hSyn-eGFP) and observed no significant effect on chloroquine-evoked scratching behavior. The reduction in chloroquine-evoked scratching in hM3Dq-injected mice after CNO administration significantly correlated with the number of hM3Dq-mCherry[+] vlPAG neurons. ($R^2 = 0.66$, $P = 0.0001$, Fig. 2d). In contrast, CNO (1 mg/kg, i.p) administration in mice expressing the inhibitory DREAAD (hM4Di) in the vlPAG, resulted in a significant increase in chloroquine evoked scratching (Fig. 2f, g), compared to mice expressing the control EGFP vector in the vlPAG. The CNO-mediated increase in chloroquine-evoked scratching in mice expressing the inhibitory DREAAD (hM4Di) in the vlPAG was dose-dependent (Supplementary Fig. 1b), and again, this increase in chloroquine-evoked scratching in hM4Di-injected mice after CNO administration demonstrated a significant correlation with the number of hM4Di-mCherry[+] vlPAG neurons. ($R^2 = 0.38$, $P = 0.01$, Fig. 2h). The results showing that targeted inhibition of resident vlPAG neurons results in increased scratching is in stark contrast to the results from the lidocaine experiment described above, where inactivation of vlPAG (including both resident neurons and fibers of passage) led to reduced scratching. These findings illustrate the importance of using cell-specific approaches, and demonstrate that activating vlPAG neurons attenuates pruritus, while inhibiting vlPAG neurons potentiates pruritic behaviors

### Itch-induced aversion is blocked by activation of vlPAG.
Itch is an unpleasant emotional experience in humans. We adapted a conditioned place aversion (CPA) paradigm to evaluate the unpleasant affective component of itch in mice (Supplementary Fig. 2a). We hypothesized that chloroquine-induced itch would produce negative reinforcement. Mice were placed in a 3-chamber CPA arena and given free access to all chambers on day one. The mice were then paired with saline injection or chloroquine injection in separate chambers distinguishable by vertical

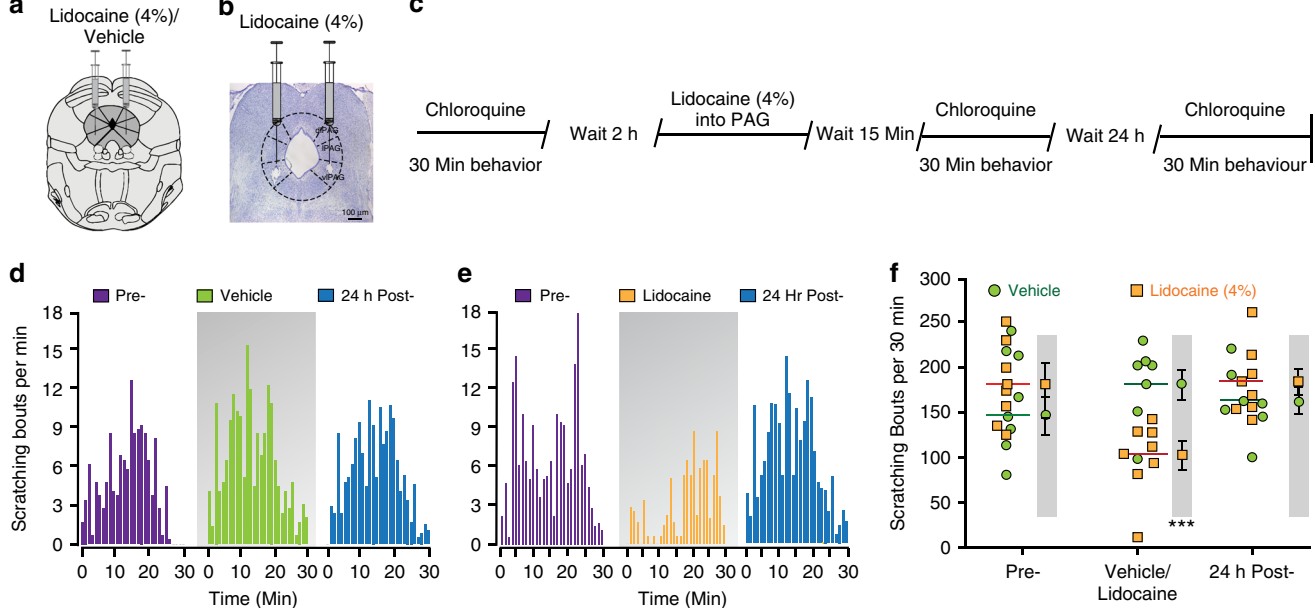

**Fig. 1** Pharmacological inactivation of the vlPAG attenutates scratching behavior. **a** Representation of bilateral placement of guide cannulae in the vlPAG in mice injected with vehicle/lidocaine. **b** Representative histological verification showing bilateral microinjection of lidocaine into the vlPAG. **c** Experimental timeline for examination of behavioral effects of bilateral microinjection of lidocaine into the vlPAG on chloroquine evoked scratching. **d, e** Histograms demonstrating the mean number of scratching bouts per minute for a 30-min period. **d** Microinjection of vehicle (0.9% saline) into vlPAG did not produce any effects on chloroquine evoked scratching when compared to baseline and 24-hour post-vehicle treatment. **e** Microinjection of lidocaine into vlPAG resulted in significant suppression of chloroquine evoked scratching when compared to baseline and 24-hour post-lidocaine treatment. **f** Summary quantification of vehicle and lidocaine microinjection effects on chloroquine evoked scratching. Data represents the total number of chloroquine evoked scratching bouts per individual mouse at prior to treatment, following injection of vehicle/lidocaine, or 24 h post-treatment with lidocaine (orange squares) or vehicle (green circles) treatment ($N = 8$ per group, two-way ANOVA, $P < 0.0001$ after lidocaine microinjection; Interaction; $F_{(2, 14)} = 26.46$, ***$P < 0.0001$. Pre vs. Post Lidocaine; $F_{(2, 14)} = 14.26$, ^^^$P < 0.0001$. Vehicle vs. Lidocaine; ###$F_{(2, 24)} = 4.71$, $P = 0.0001$). Values in the gray shaded areas represent the mean ± SEM of all animals, for which individual values are shown in the un-shaded areas (horizontal lines represent means). Source data are available as a Source Data file

or horizontal stripes for three subsequent days. Twenty-four hours after the last pairing, mice were given free access to all chambers and the time spent in each chamber was recorded. Consistent with our hypothesis, mice exhibited robust aversion to the chamber paired with chloroquine injection compared to the chamber paired with saline injection (Supplementary Fig. 2b & c). CPA to chloroquine was dose dependent; 50 µg/10 µl chloroquine did not show CPA while doses of 200 µg and 400 µg/10 µl produced significant CPA (Supplementary Fig. 2c). All groups spent an equivalent time in the neutral chamber. These results suggest that chloroquine produces reliable, robust place aversion, consistent with a recent report[34].

Having demonstrated that chemogenetic activation of vlPAG neurons suppresses chloroquine-induced scratching, we next asked whether chemogenetic activation of vlPAG neurons would abolish place aversion to chloroquine. We transduced vlPAG neurons bilaterally with AAV8 vectors carrying either the stimulatory DREADD (hM3Dq) fused with mCherry or control AAV8 EGFP. After free access to both chambers on day 1, mice were conditioned for 3 days, by administering either saline or chloroquine. On conditioning days, mice were pre-treated with either saline or CNO (i.p) 1 h before being paired in the specified chambers with saline or chloroquine injection (Fig. 3a). When tested on day 5 after conditioning, animals expressing the control EGFP construct demonstrated robust place aversion to chloroquine + CNO, similar to what was seen in naïve mice (Fig. 3b, c, $n = 13$, $t$ test, $p = 0.0219$). In contrast, animals expressing the stimulatory DREADD hM3Dq in the vlPAG showed no aversion to the chloroquine + CNO-paired chamber, (Fig. 3d, e, $n = 13$, $t$ test, $p = 0.9859$). These results demonstrate that activation of

hM3Dq in vlPAG neurons blocks place aversion to chloroquine and suggest that activation of vlPAG neuronal circuitry attenuates the unpleasant affective component of itch.

**Ca2+ dynamics of vlPAG cell-types in itch processing**. The results above show that excitation of vlPAG neurons leads to suppression of chloroquine-evoked scratching, whereas inhibition of vlPAG neuronal activity potentiates chloroquine-evoked scratching. However, the vlPAG is composed of both inhibitory GABAergic and excitatory glutamatergic neurons, which are non-overlapping populations[35–37]. The chemogenetic manipulations of vlPAG neuronal activity in the prior experiments do not distinguish between these two populations. To investigate the potential for differential roles of vlPAG GABAergic and glutamatergic neurons in the regulation of itch behaviors, we used fiber photometry[38] to measure bulk Ca$^{2+}$ dynamics in a cell type-specific manner in awake, behaving mice. By expressing the genetically encoded calcium indicator, GCaMP6s, in GABAergic (Vgat-IRES-Cre) or glutamatergic (Vglut2-IRES-Cre) neurons, we can measure neuronal activity-dependent population fluorescence changes in specific cell types within the vlPAG while mice are engaged in pruritogen-induced scratching. We virally expressed GCaMP6s in Vgat$^+$ or Vglut2$^+$ vlPAG neurons by injecting Cre-dependent AAV (AAV-DJ EF1a-DIO-GCaMP6s) into the vlPAG of Vgat-Cre or Vglut2-Cre mice. An optical fiber was implanted unilaterally above the vlPAG to measure changes in Ca$^{2+}$ dynamics during chloroquine-evoked scratching (Fig. 4a, b). Mice expressing GCaMP6s in either Vgat or Vglut2 neurons were acclimated to a behavioral chamber and injected

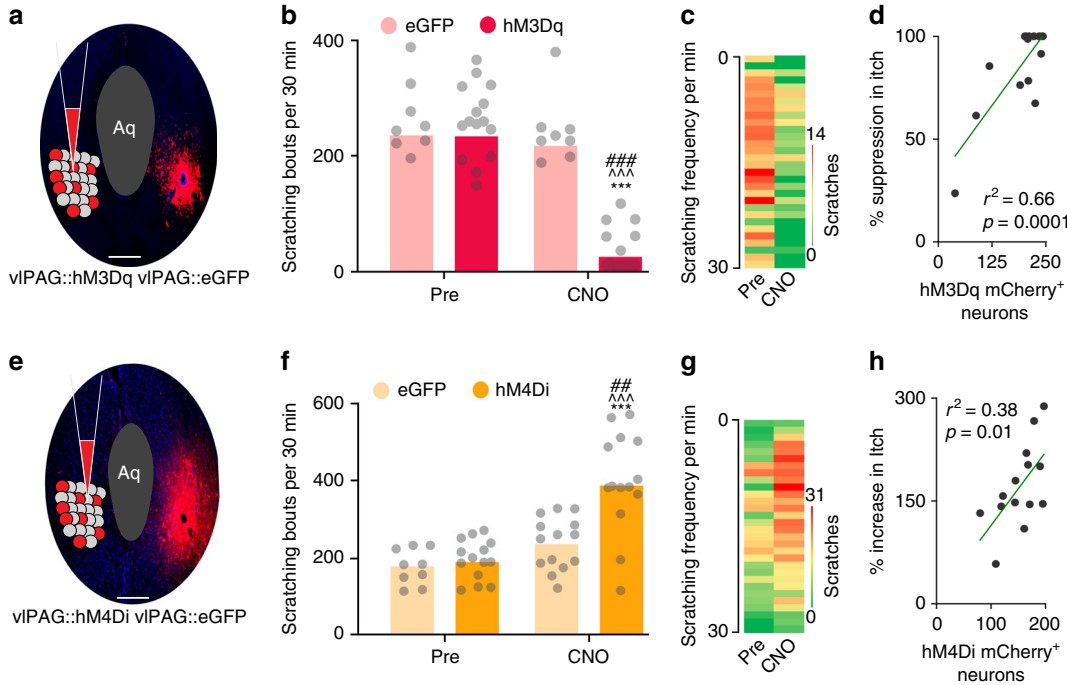

**Fig. 2** Chemogenetic manipulation of vlPAG neurons can modulates itch. **a** Schematic and representative coronal section image showing viral targeting of AAV8 hM3Dq–mCherry transgenes bilaterally injected into the vlPAG, scale bar represents 375 μM. **b** CNO (1 mg/kg, i.p) administration resulted in a significant reduction in chloroquine evoked scratching in hM3Dq injected mice. Relative to pre-treatment control values, CNO (1 mg/kg, i.p) did not have any effects on the number of scratches induced by chloroquine injection in mice expressing the control eGFP construct Interaction; $F_{(1, 21)} = 25.24$, ***$P < 0.0001$. Pre vs. CNO; $F_{(1, 21)} = 37.49$, ^^^$P < 0.0001$. eGFP vs. hM3Dq; $F_{(1, 21)} = 25.65$, ###$P < 0.0001$. $N = 15$ for hM3Dq and $N = 8$ for eGFP group. Two-way ANOVA. **c** Heat map showing per-minute averaged chloroquine-evoked scratching bouts pre-CNO and post-CNO in mice expressing hM3Dq in the vlPAG. **d** Reduction in chloroquine evoked scratching in hM3Dq injected mice after CNO administration is directly correlated with number of hM3Dq-mCherry$^+$ vlPAG neurons. $R^2 = 0.66$, $P = 0.0001$. **e** Illustration and representative image of coronal section containing vlPAG demonstrating restricted viral expression following microinjection of the AAV8 hM4Di-mCherry into the vlPAG, scale bar represents 305 μM. **f** CNO (1 mg/kg, i.p) administration resulted in a significant increase in chloroquine evoked scratching in hM4Di injected mice but not in control mice that express eGFP construct. Interaction; $F_{(1, 23)} = 17.03$, ***$P = 0.0004$. Pre vs. CNO; $F_{(1, 23)} = 22.77$, ^^^$P < 0.0001$. eGFP vs. hM4Di; $F_{(1, 23)} = 9.915$, ##$P = 0.0045$. $N = 14$ for hM4Di and $N = 11$ for eGFP control. Two-way ANOVA. **g** Heat map showing per-minute averaged chloroquine-evoked scratching bouts pre-CNO and post-CNO in mice expressing hM4Di in the vlPAG. **h** Increase in chloroquine evoked scratching in hM4Di injected mice after CNO administration is directly correlated with number of hM4Di-mCherry$^+$ vlPAG neurons. $R^2 = 0.38$, $P = 0.01$. All values are mean ± SEM. Source data are available as a Source Data file

intradermally with chloroquine. We found that vlPAG Vgat+ and Vglut2+ neurons showed opposing responses during chloroquine-evoked scratching. Vgat$^+$ vlPAG neurons showed significant decreases of Ca$^{2+}$ transients during chloroquine-evoked scratching bouts. These decreases in Ca$^{2+}$ signals began immediately following the initiation of scratching bouts and lasted for 30 s on average (Fig. 4c–e, h). Vglut2$^+$ vlPAG neurons showed the opposite response, wherein chloroquine-evoked scratching bouts resulted in significant and long-lasting increases in Ca$^{2+}$ transients, beginning at the onset of the scratching behaviors (Fig. 4c, f, g, i). We did not observe similar robust changes in Ca$^{2+}$ dynamics during spontaneous scratching bouts in the Vgat+ neurons prior to chloroquine evoked scratching (Supplementary Fig. 3). The decrease in the Vgat Ca$^{2+}$ activity was significantly greater during chloroquine-evoked scratching events compared to these spontaneous scratches recorded prior to chloroquine injection. We tested the itch-specific nature of these changes in vlPAG calcium dynamics by quantifying changes in fluorescence of the same neuronal populations during grooming or wiping behaviors. At the onset of grooming or wiping, Ca$^{2+}$ dynamics were variable but not significantly changed, suggesting that the fluorescence changes described above are functionally related to chloroquine-induced scratching (Fig. 4j, i). To further verify that changes observed in Vgat+ and Vglut2+ vlPAG neurons are correlated with chloroquine-evoked scratching

behaviors, we generated randomly assigned arbitrary bout events and aligned Ca$^{2+}$ transients to these arbitrary bouts. Ca$^{2+}$ dynamics of Vgat+ and Vglut2+ vlPAG neurons were not significantly correlated with these arbitrarily created bouts (Fig. 4k, m). Altogether these data suggest that vlPAG Vgat$^+$ and Vglut2$^+$ neurons respond differently during chloroquine-evoked scratching. Our results capture the rapid changes in activity that these neurons undergo in response to chloroquine-evoked scratching and suggest that a cell type-specific approach is needed to understand the role of vlPAG neurons in sensory and affective aspects of itch.

**vlPAG GABAergic neurons modulate acute and chronic itch.** The fiber photometry studies described above show a decrease in activity of Vgat$^+$ neurons during chloroquine-evoked scratching, suggesting a potential role for reduced GABAergic transmission in the vlPAG in itch processing. To test the hypothesis that altered vlPAG GABAergic activity regulates scratching behaviors, we introduced Cre-dependent viral constructs containing either DREADDs fused to mCherry or control construct lacking the DREADD (Syn-mCh) into the vlPAG of Vgat-IRES-Cre mice (Fig. 5a). We have recently shown that CNO reliably modulates the activity of vlPAG GABAergic neurons expressing DREADDs[36]. We selectively activated vlPAG GABAergic

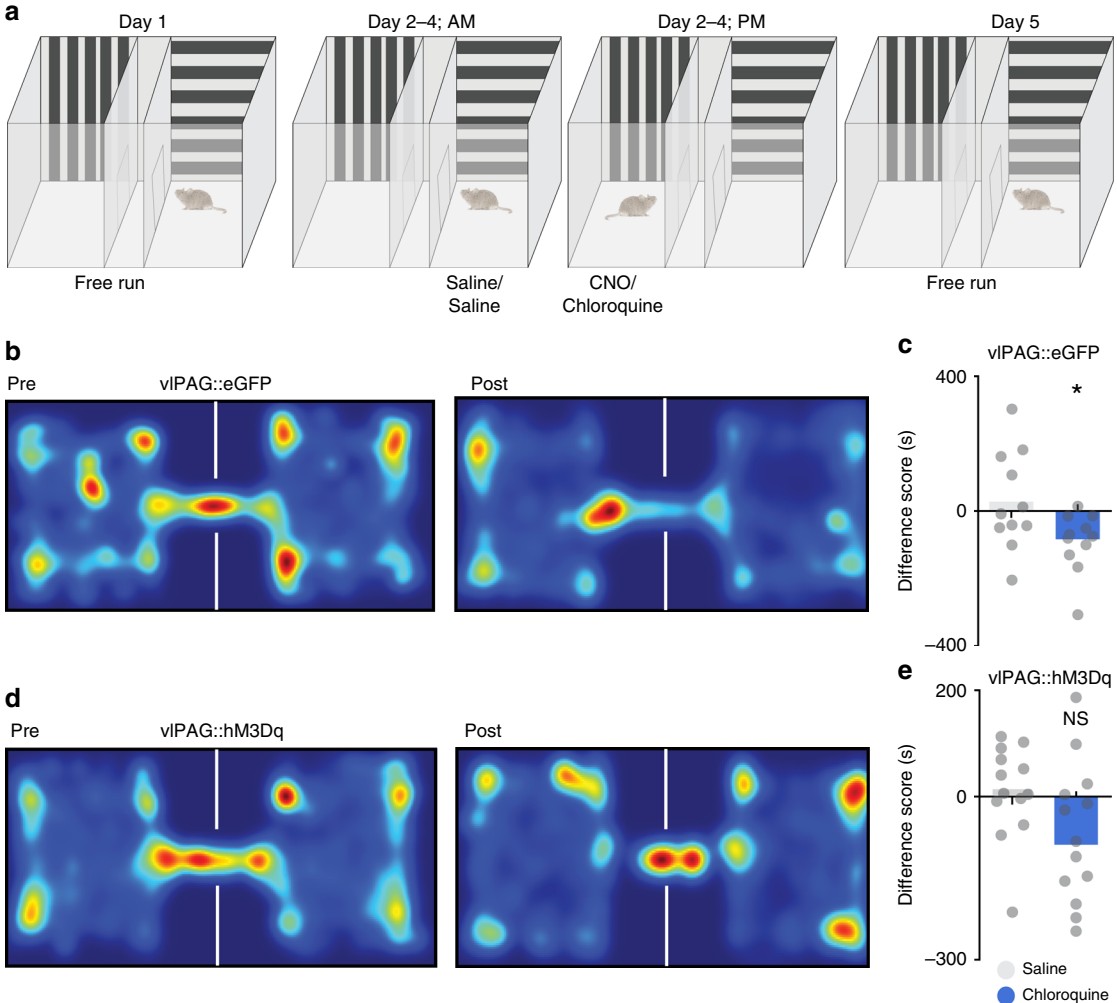

**Fig. 3** Chemogenetic activation of vlPAG neurons can block aversive component of itch. **a** Schematic of conditioned place aversion experimental design in AAV8 hM3Dq–mCherry and AAV8–eGFP injected animals. Mice received treatment with either i.p. saline or CNO 90 min before pairing the mice in the chambers with saline or chloroquine, respectively. **b** Representative heat map showing spatial location of a control mouse injected with AAV8–EGFP, pre and post chloroquine conditioning. **c** After CNO (0.5 mg/kg, i.p) administration in vlPAG::eGFP mice, difference scores indicate mice spent significantly less time in the chloroquine paired chamber compared to saline paired chamber ($N = 13$, $t$ test, $t = 2.128$ df = 24, $*P = 0.0219$). **d** Representative heat map showing spatial location of a mouse injected with AAV8–hM3Dq, pre and post chloroquine conditioning. **e** Activation of vlPAG::hM3Dq neurons with CNO (0.5 mg/kg, i.p) before mice were conditioned with chloroquine resulted in disruption of CPA to chloroquine ($N = 13$, $t$ test, $t = 0.01788$ df = 24 $P = 0.9859$). All values are mean ± SEM. Source data are available as a Source Data file

neurons via expression of the Cre-dependent excitatory DREADD hM3Dq in the vlPAG (Fig. 5a). In vivo CNO-dependent (5 mg/kg, i.p.) activation of vlPAG Vgat$^+$ neurons resulted in a significant decrease in chloroquine-evoked scratching compared to baseline (Fig. 5b, c). The reduction in chloroquine evoked scratching in hM3Dq-injected mice after CNO administration is dose dependent (Supplementary Fig. 1c) and significantly correlated with the number of hM3Dq-mCherry$^+$ vlPAG neurons ($R^2 = 0.68$, $P = 0.0004$, Fig. 5d). These results suggest that chloroquine-induced scratching can be suppressed by the activation of Gq signaling in GABAergic vlPAG neurons. We next asked what would happen to chloroquine-evoked scratching if we decreased the activity of vlPAG Vgat$^+$ neurons. Three weeks after viral infection of vlPAG neurons with a Cre-dependent viral vector for the inhibitory DREADD hM4Di, we observed restricted expression of hM4Di-mcherry in GABAergic neurons within the vlPAG of Vgat-Cre mice (Fig. 5a). Inhibition of vlPAG GABAergic neurons with CNO (5 mg/kg; i.p) resulted in a robust increase in chloroquine-evoked scratching compared to baseline (Fig. 5e, f). Potentiation of chloroquine-evoked scratching in

hM4Di-injected mice after CNO administration is dose dependent (Supplementary Fig. 1d) but not significantly correlated with the number of hM4Di-mCherry$^+$ vlPAG neurons ($R^2 = 0.1$, $P = 0.5$, Fig. 5g). Mice expressing the control mCherry vector in the vlPAG did not show any changes in scratching behavior following CNO administration.

To investigate whether activation of vlPAG GABAergic neurons can attenuate chronic itch, we used a dry skin model of itch. Chronic spontaneous scratching develops following daily topical application of acetone/ether/water (AEW) for 7 days[39–41]. Seven days after AEW treatment, Vgat::mCh-injected and Vgat::hM3Dq-injected mice showed robust spontaneous scratching. On day 8, chemogenetic activation of vlPAG GABAergic neurons with CNO (5 mg/kg, i.p.) resulted in a significant reduction of spontaneous scratching behavior produced by AEW treatment (Fig. 5h). In Vgat::mCherry mice, CNO administration did not affect spontaneous scratching behavior produced by AEW treatment.

We next tested whether chemogenetic activation of vlPAG GABAergic neurons would block the aversive component of

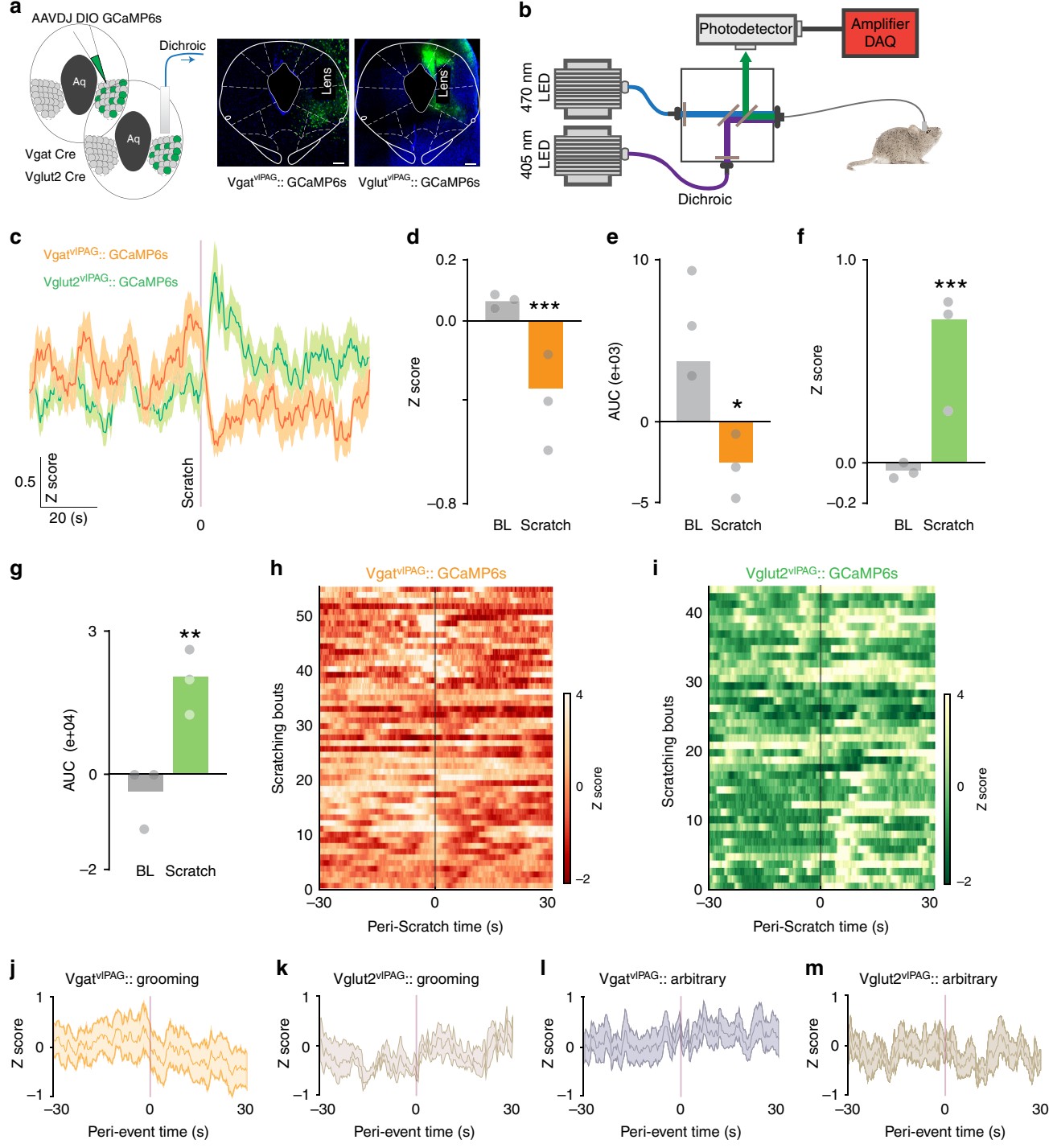

chloroquine-induced itch. We performed CPA to chloroquine in Vgat::hM3Dq and Vgat::mCherry mice. During conditioning, mice were pre-treated with either saline or CNO (i.p) 1 h before being placed in the saline or chloroquine paired chamber (Fig. 6a). When allowed free access to both chambers following conditioning, Vgat::hM3Dq-injected mice spent similar time in both chambers whereas Vgat::mCherry-injected controls spent significantly less time in the chamber that was paired with chloroquine (Fig. 6b, c, d and e). Neither Vgat::hM3Dq nor Vgat::mCherry showed any change in chamber preference from their initial exploration. Neither group showed altered locomotor activity following CNO administration. These results demonstrate that activation of vlPAG GABAergic neurons attenuates place

aversion to chloroquine and suggest that these neurons modulate the aversive component of itch.

## vlPAG glutamatergic neurons modulate acute and chronic itch.
The increases in vlPAG Vglut2$^+$ neuronal activity observed during scratching (fiber photometry recordings above) suggest a potential causal relationship between activity of these neurons and chloroquine-evoked scratching behaviors (Fig. 4c, i). To test this hypothesis, we expressed Cre-dependent hM3Dq-mCherry or a control virus lacking the DREADD (Syn-mCh) selectively in glutamatergic (Vglut2-expressing) vlPAG neurons (Fig. 7a) and asked if increasing the activity of Vglut2 neurons could augment

**Fig. 4** Ca2 + dynamics of vlPAG cell-types in itch processing. **a** Schematic and representative coronal section image showing viral targeting of AAVDJ GCaMP6s injected into the vlPAG in the Vgat and Vglut2 Cre mice. **b** Illustration of fiber photometry setup. **c** Averaged GCAMP6s fluorescence signal of Vgat[+ve] and Vglut[+ve] vlPAG neurons showing changes in fluorescence after initiation of scratching bout. Trace plotted as mean ± SEM (shaded trace), dark shaded line indicates scratching bout. **d** Averaged Z score for chloroquine-evoked scratching resulted in significant decrease in the vlPAG GCAMP6s fluorescence in the Vgat[+ve] neurons. ($N = 3$ mice, t test, $t = 10.07$, df = 4, ***$P < 0.0001$). **e** Area under the curve shows significant decrease in the vlPAG GCAMP6s fluorescence after chloroquine-evoked scratching in the Vgat[+ve] neurons. ($N = 3$ mice, t test, $t = 2.435$, df = 4, *$P < 0.05$. **f** Averaged Z score for chloroquine-evoked scratching resulted in significant increase in the vlPAG GCAMP6s fluorescence in the Vglut[+ve] neurons. ($N = 3$ mice, t test, $t = 5.875$, df = 4, **$P < 0.0001$). **g** Area under the curve shows significant increase in the vlPAG GCAMP6s fluorescence after chloroquine-evoked scratching in the Vglut[+ve] neurons. ($N = 3$ mice, t test, $t = 5.964$, df = 4, **$P < 0.01$. (**h**) Heat map of Ca2+ dynamics of Vgat[+ve] vlPAG neurons pre and post chloroquine evoked scratching bouts. Dark shaded line indicates scratching bout. **i** Heat map of Ca2+ dynamics of Vglut[+ve] vlPAG neurons pre and post chloroquine evoked scratching bouts. Dark shaded line indicates scratching bout. **j** Averaged GCAMP6s fluorescence signal of Vgat[+ve] vlPAG neurons after initiation of grooming bout. Trace plotted as mean ± SEM (shaded trace). **k** Averaged GCAMP6s fluorescence signal of Vglut[+ve] vlPAG neurons after initiation of grooming bout. Trace plotted as mean ± SEM (shaded trace), dark shaded line indicates grooming bout. **l** Averaged GCAMP6s fluorescence signal of Vgat[+ve] vlPAG neurons for arbitrarily generated bouts. Trace plotted as mean ± SEM (shaded trace). **m** Averaged GCAMP6s fluorescence signal of Vglut[+ve] vlPAG neurons for arbitrarily generated bouts. Trace plotted as mean ± SEM (shaded trace), dark shaded line indicates arbitrary bout. All values are mean ± SEM. $P < 0.0001$. Scale bars represents 250 μM. Source data are available as a Source Data file

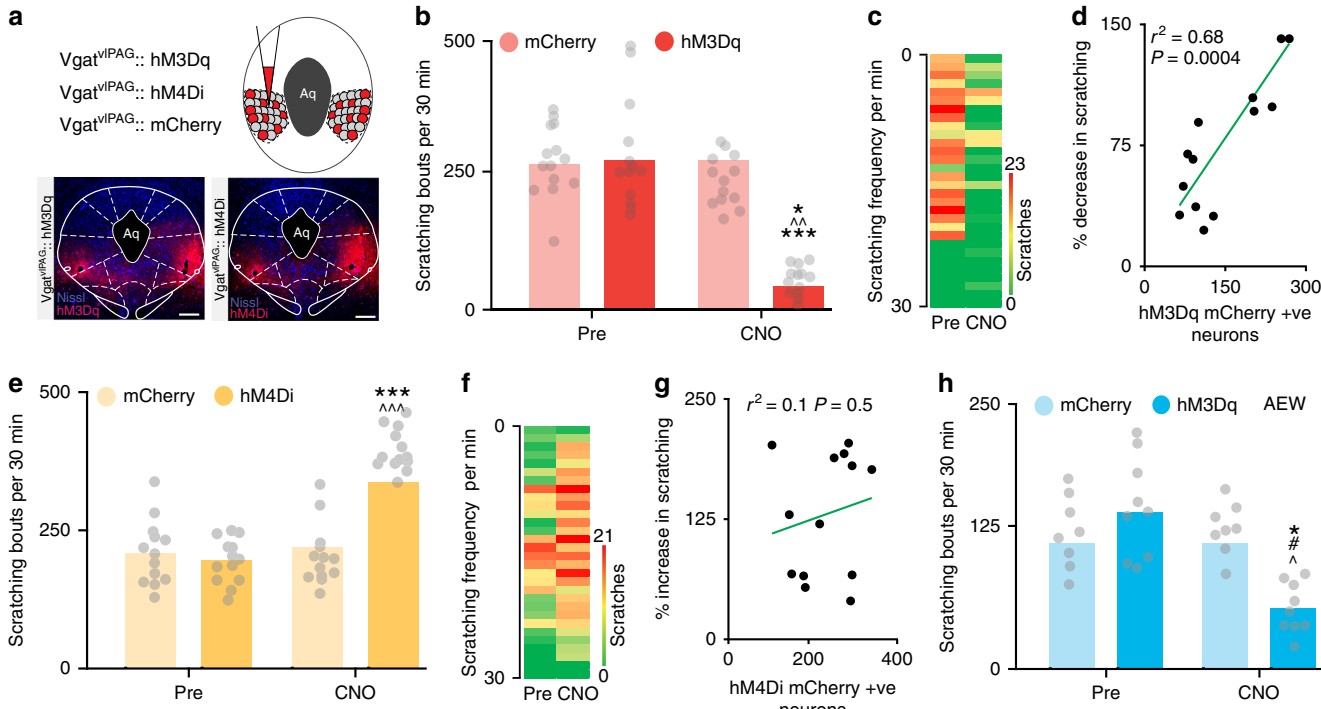

**Fig. 5** vlPAG GABAergic neuronal activity modulates sensory component of itch. **a** Viral targeting of AAV5-hSyn-DIO-hM3Dq–mCherry, AAV5-hSyn-DIO-hM4Di–mCherry and AAV5-hSyn-DIO-mCherry bilaterally into the vlPAG of Vgat Cre mice. Representative vlPAG sections showing restricted viral expression of hM3Dq–mCherry and hM4Di–mCherry in the vlPAG of Vgat Cre mice. **b** CNO (5 mg/kg, i.p) had no significant effect on chloroquine evoked scratching in Vgat Cre mice expressing the control mCherry construct in the vlPAG. However, CNO (5 mg/kg, i.p) administration resulted in significant reduction in chloroquine evoked scratching in mice that express hM3Dq in the vlPAG of Vgat Cre mice. Interaction; F (1, 24) = 21.14, ^^$P < 0.001$. Pre vs. CNO; F (1, 24) = 7.531, *$P = 0.0113$. mCherry vs. hM3Dq; F (1, 24) = 39.20, ***$P < 0.0001$. $N = 13$/group. Two-way ANOVA. **c** Heat map showing averaged chloroquine-evoked scratching bouts pre-CNO and post-CNO. **d** Chloroquine evoked scratching in hM3Dq injected mice after CNO administration is directly correlated with number of hM3Dq-mCherry[+] neurons. $R^2 = 0.68$, $P = 0.0004$. **e** Inhibition of vlPAG neurons by CNO (5 mg/kg, i.p) administration resulted in a significant increase in chloroquine evoked scratching in Vgat::hM4Di mice compared to baselines while control mCherry injected mice in the vlPAG show no difference in chloroquine evoked scratching after CNO (5 mg/kg, i.p) administration. Interaction; F (1, 24) = 32.55, ^^^$P < 0.0001$. Pre vs. CNO; F (1, 24) = 27.60, ###$P < 0.0001$. mCherry vs. hM4Di; F (1, 24) = 40.65, ***$P < 0.0001$. $N = 13$/group. Two-way ANOVA. **f** Heat map showing averaged chloroquine-evoked scratching bouts pre-CNO and post-CNO in mice expressing hM4Di. **g** Chloroquine evoked scratching in hM4Di injected mice after CNO administration is not correlated with number of hM4Di-mCherry[+] vlPAG in the Vgat Cre mice. $R^2 = 0.1$, $P = 0.5$. **h** Activation of vlPAG GABAergic neurons with CNO (5 mg/kg, i.p.) resulted in a significant reduction of spontaneous scratching behavior produced by AEW treatment. CNO administration did not affect spontaneous scratching behavior produced by AEW treatment in control mCherry injected mice. Interaction; F (1, 14) = 9.056, ***$P = 0.0094$. Pre vs. CNO; F (1, 14) = 13.58, ^^$P = 0.0024$. $N = 8$/group. Two-way ANOVA. All values are mean ± SEM. Source data are available as a Source Data file

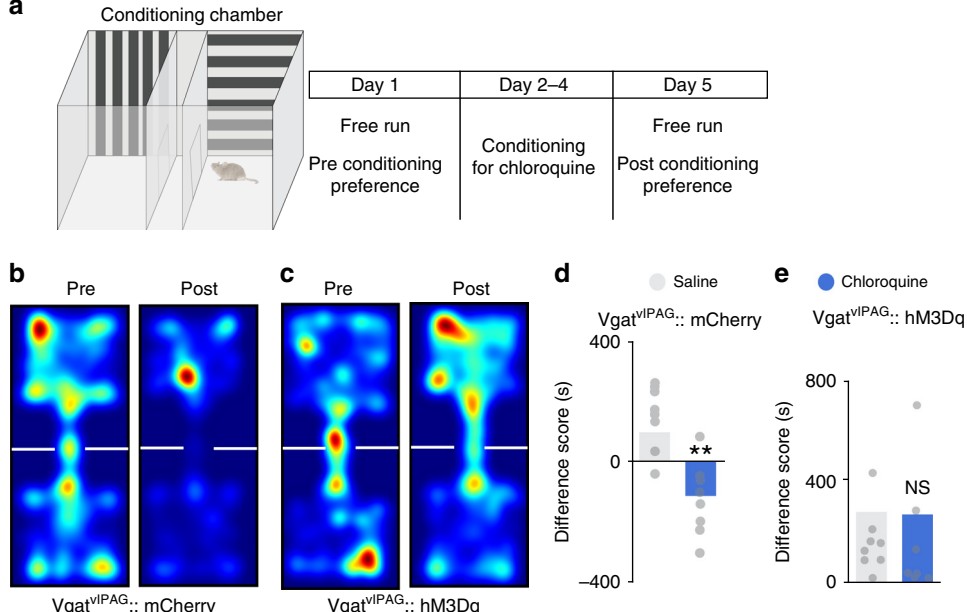

**Fig. 6** vlPAG GABAergic neuronal activity modulates aversive component of itch. **a** Schematic of conditioned place aversion experimental design indicating saline-paired and chloroquine-paired chambers and the timing of each session in Vgat::hM3Dq and Vgat::mCherry mice. **b** Representative heat map showing location of a control mouse injected with mCherry in the vlPAG of Vgat Cre mice, pre and post chloroquine conditioning. (c Representative heat map showing location of a mouse injected with hM3Dq in the vlPAG of Vgat Cre mice, pre and post chloroquine conditioning. **d** CNO (5 mg/kg, i.p) administration in vlPAG::mCherry mice did not have any effect on the time spent in the chloroquine paired chamber compared to saline paired chamber ($N = 8$, t test, $t = 3.920$, df $= 13$, **$P = 0.0018$). **e** Activation of Vgat::hM3Dq neurons in the vlPAG with CNO (5 mg/kg, i.p) before mice were conditioned with chloroquine resulted in disruption of CPA to chloroquine ($N = 7$, t test, $t = 0.05297$, df $= 12$ $P = 0.9586$). All values are mean ± SEM. Scale bars indicate 155 μM. Source data are available as a Source Data file

chloroquine-evoked scratching behavior. It is important to note that we recently demonstrated that CNO reliably modulates the activity of vlPAG glutamatergic neurons expressing DREADDs[36]. Three weeks after DREADD injection, robust viral expression was observed in the vlPAG (Fig. 7a). Activation of vlPAG Vglut2::hM3Dq neurons via CNO (5 mg/kg, i.p.) led to a profound increase in chloroquine-evoked scratching compared to baseline (Fig. 7b, c). CNO administration in Vglut2::mCherry mice had no effect on chloroquine-evoked scratching, indicating that elevated glutamatergic activity in the vlPAG is sufficient to enhance chloroquine-evoked scratching. The increase in chloroquine evoked scratching in hM3Dq-injected mice after CNO administration is dose dependent (Supplementary Fig. 1e) and significantly correlated with the number of hM3Dq-mCherry$^+$ vlPAG neurons ($R^2 = 0.39$, $P = 0.01$, Fig. 7d).

Next, we assessed whether endogenous activity of vlPAG Vglut2 neurons is necessary for regulating itch processing. Inhibition of Vglut2 + vlPAG neurons in Vglut2::hM4Di mice with CNO (5 mg/kg; i.p) resulted in a significant decrease in chloroquine-evoked scratching compared to baseline (Fig. 7e, f). The CNO-mediated decrease in chloroquine-evoked scratching is dose dependent (Supplementary Fig. 1f). Mice expressing the control mCherry vector in vlPAG glutamatergic neurons show no significant change in scratching behavior upon CNO administration. These decreases also significantly correlate with the number of hM4Di-mCherry$^+$ vlPAG neurons in Vglut2 Cre mice ($R^2 = 0.45$, $P = 0.005$, Fig. 7g).

We further analyzed whether inhibition of vlPAG glutamatergic neurons can affect scratching behaviors in the AEW dry skin model of chronic itch. Vglut2::mCh-injected and Vglut2::hM4Di-injected mice showed robust spontaneous scratching seven days after AEW treatment. Chemogenetic inhibition of vlPAG glutamatergic neurons (Vglut2::hM4Di) with CNO (5 mg/kg,

i.p.) resulted in a significant reduction of spontaneous scratching behavior produced by dry skin (Fig. 7h). CNO administration in Vglut2::mCherry mice did not lead to changes in spontaneous scratching behavior produced by dry skin. These results demonstrate that activity of vlPAG glutamatergic neurons is necessary for acute and chronic itch processing.

To determine if chemogenetic inhibition of vlPAG glutamatergic neurons can block the aversive component of chloroquine-induced itch, we performed CPA to chloroquine in Vglut2::hM4Di and Vglut2::mCherry mice (Fig. 8a). In contrast to what we observed following modulation of vlPAG GABAergic neurons, modulation of the activity of glutamatergic vlPAG neurons had no detectable effect on place aversion induced by chloroquine. Thus, when allowed free access to both chambers following conditioning, both Vglut2::hM3Dq and Vglut2::mCherry injected mice spent significantly less time in the chloroquine-paired chamber, suggesting that place aversion was not suppressed by activation of vlPAG glutamatergic neurons. Neither Vglut2::mCherry mice (Fig. 8b, d) nor Vglut2::hM3Dq (Fig. 8c, e) showed any change in behavior from their initial exploration and neither group showed altered locomotor activity following CNO administration. These results demonstrate that inhibition of vlPAG glutamatergic neurons did not prevent place aversion to chloroquine and suggest that vlPAG glutamatergic neurons do not contribute to the aversive component of itch.

**PAG neurons can modulate spinal pruritic processing.** To determine whether global manipulation of PAG neurons modulates pruritic processing via descending control at the level of the spinal cord, we performed activity-dependent mapping studies using cFos labeling (a surrogate marker for neuronal activity). We found chloroquine injection induced cFos expression in the spinal

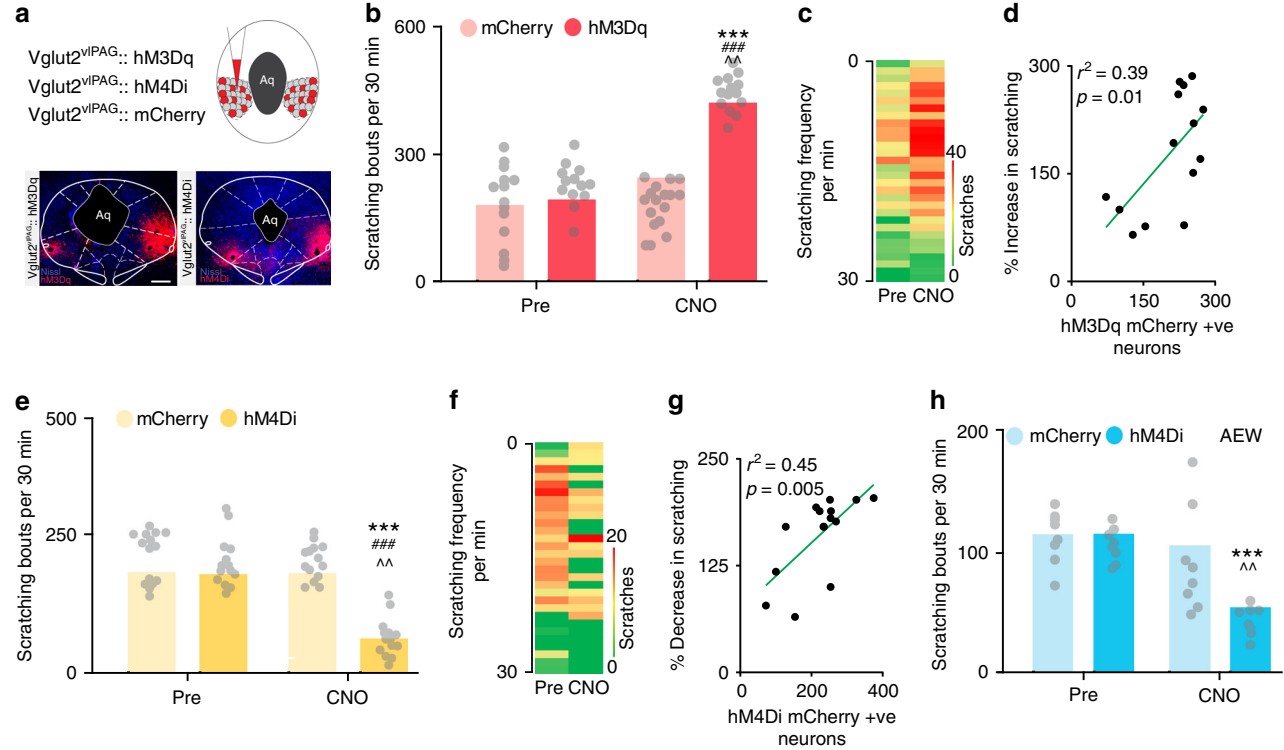

**Fig. 7** vlPAG glutamatergic neurons bidirectionally modulates itch. **a** Illustration showing viral targeting strategy of AAV5-hSyn-DIO-hM3Dq–mCherry, AAV5-hSyn-DIO-hM4Di–mCherry and AAV5-hSyn-DIO-mCherry bilaterally injected into the vlPAG of Vglut2 Cre mice. Representative images of coronal sections containing vlPAG showing restricted viral expression following microinjection of AAV5-hSyn-DIO-hM3Dq–mCherry and AAV5-hSyn-DIO-hM4Di–mCherry in the vlPAG of Vglut2 Cre mice. **b** CNO (5 mg/kg, i.p) had no significant effect on chloroquine evoked scratching in Vglut2 Cre mice expressing the control mCherry construct in the vlPAG. However, CNO (5 mg/kg, i.p) administration resulted in significant increase in chloroquine evoked scratching in mice that express hM3Dq in the vlPAG of Vglut2 Cre mice. Interaction; $F_{(1, 26)} = 40.17$, ***$P < 0.0001$. Pre vs. CNO; $F_{(1, 26)} = 91.06$, ^^$P = 0.0001$. mCherry vs. hM3Dq; $F_{(1, 26)} = 99.35$, ###$P < 0.0001$. $N = 14$/group. Two-way ANOVA. **c** Heat map showing per-minute averaged chloroquine-evoked scratching bouts pre-CNO and post-CNO. **d** Chloroquine evoked scratching in hM3Dq injected mice after CNO administration is directly correlated with number of hM3Dq-mCherry+ vlPAG neurons. $R^2 = 0.39$, $P = 0.01$. **e** Inhibition of vlPAG neurons by CNO (5 mg/kg, i.p) administration resulted in a significant decrease in chloroquine evoked scratching in Vglut2::hM4Di mice compared to baselines while control mCherry injected mice in the vlPAG show no difference in chloroquine evoked scratching after CNO (5 mg/kg, i.p) administration. Interaction; $F_{(1, 27)} = 11.41$, ^^$P = 0.0025$. Pre vs. CNO; $F_{(1, 27)} = 18.35$, ***$P = 0.0003$. mCherry vs. hM4Di; $F_{(1, 27)} = 24.26$, ###$P < 0.0001$. $N = 14$/group. Two-way ANOVA. **f** Heat map showing per-minute averaged chloroquine-evoked scratching bouts pre-CNO and post-CNO. **g** Chloroquine evoked scratching in hM4Di injected mice after CNO administration is directly correlated with number of hM4Di-mCherry+ vlPAG neurons in Vglut2 Cre mice. $R^2 = 0.45$, $P = 0.005$. **h** Chemogenetic inhibition of vlPAG glutamatergic neurons with CNO (5 mg/kg, i.p.) resulted in a significant reduction of spontaneous scratching behavior produced by AEW treatment. CNO administration did not affect spontaneous scratching behavior produced by AEW treatment in control mCherry injected mice. Interaction; $F_{(1, 16)} = 3.816$, *$P = 0.0685$. Pre vs. CNO; $F_{(1, 16)} = 7.029$, ^$P = 0.0174$. mCherry vs. hM3Dq $F_{(1, 16)} = 3.341$, $P = 0.0863$. $N = 8$/group. Two-way ANOVA. All values are mean ± SEM. Source data are available as a Source Data file

cord, and that global chemogenetic activation of PAG neurons suppressed this chloroquine-induced induction of spinal cFos expression (Fig. 9b, d), whereas global chemogenetic inhibition of PAG neurons lead to increases in the spinal cFos to chloroquine stimuli (Fig. 9c, d). These results suggest that PAG neurons may modulate spinal pruritic processing by descending projections to the spinal cord via the rostral ventromedial medulla (RVM), as suggested by recent work[42].

## Discussion

More than 70 years ago, the brainstem was shown to play a crucial role in itch modulation[27]. Recent studies have acknowledged the presence of defined neural circuits underlying itch processing in the brainstem, supporting this hypothesis[11,12,43]. It has been demonstrated that electrical stimulation of the PAG suppresses responsiveness of spinal cord neurons to histamine[28], but the neural subtypes involved in modulating itch processing have never been identified. Our initial experiment to test for a role of the vlPAG in itch processing in mice utilized local

pharmacologic blockade of neuronal activity (lidocaine), and supported the above studies suggesting a role for the PAG in itch modulation. This approach lacks cell-type specificity, and additionally inhibits axons coursing through or nearby the injections site. Our findings using global (not cell-type-specific) DREADD manipulation of vlPAG neurons demonstrate that activation of vlPAG neurons results in robust suppression of chloroquine-evoked scratching, while inhibition of vlPAG neurons results in profound potentiation of chloroquine-evoked scratching. This approach eliminated the possible stimulation or inhibition of axons coursing through the PAG that confounds studies utilizing either electrical stimulation-induced activation or silencing via cooling or local anesthetic administration. However, the experiments that followed demonstrate diverse roles for different cell types in the vlPAG, in a manner not easily predicted from global electrical, pharmacologic, or chemogenetic approaches described here and previously.

The PAG comprises a diverse group of neuronal subpopulations that express a variety of neurotransmitters and neuropeptides[44–46].

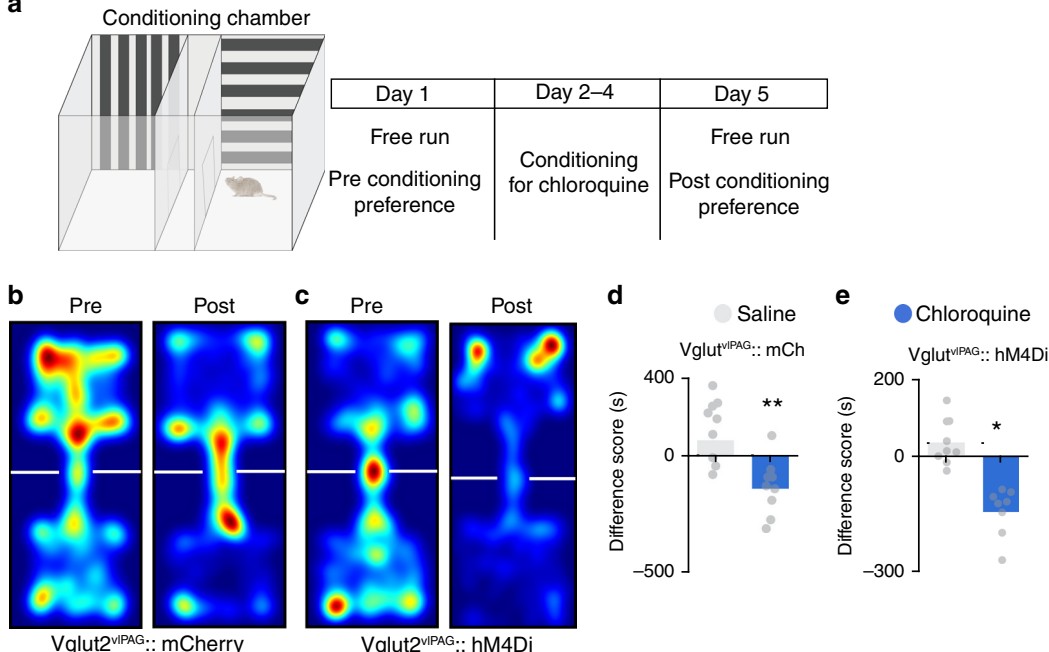

**Fig. 8** vlPAG glutamatergic neurons do not mediate aversive component of itch. **a** Schematic of conditioned place aversion experimental design indicating saline-paired and chloroquine-paired chambers and the timing of each session in Vglut2::hM4Di and Vglut2::mCherry mice. **b** Representative heat map showing location of a control mouse injected with mCherry in the vlPAG of Vglut2 Cre mice, pre-chloroquine and post-chloroquine conditioning. **c** Representative heat map showing location of a mouse injected with hM4Di in the vlPAG of Vglut2 Cre mice, pre-chloroquine and post-chloroquine conditioning. **d** CNO (5 mg/kg, i.p) administration in vlPAG::mCherry mice did not have any effect on the time spent in the chloroquine paired chamber compared to saline paired chamber ($N = 9$, $t$ test, $t = 3.435$, df $= 16$, **$P = 0.0034$). **e** Inhibition of Vglut2::hM4Di neurons in the vlPAG with CNO (5 mg/kg, i.p) before mice were conditioned with chloroquine did not affect CPA to chloroquine ($N = 8$, $t$ test, $t = 3.088$, df $= 14$, *$P = 0.0080$). All values are mean ± SEM. Scale bars represent 150 μM. Source data are available as a Source Data file

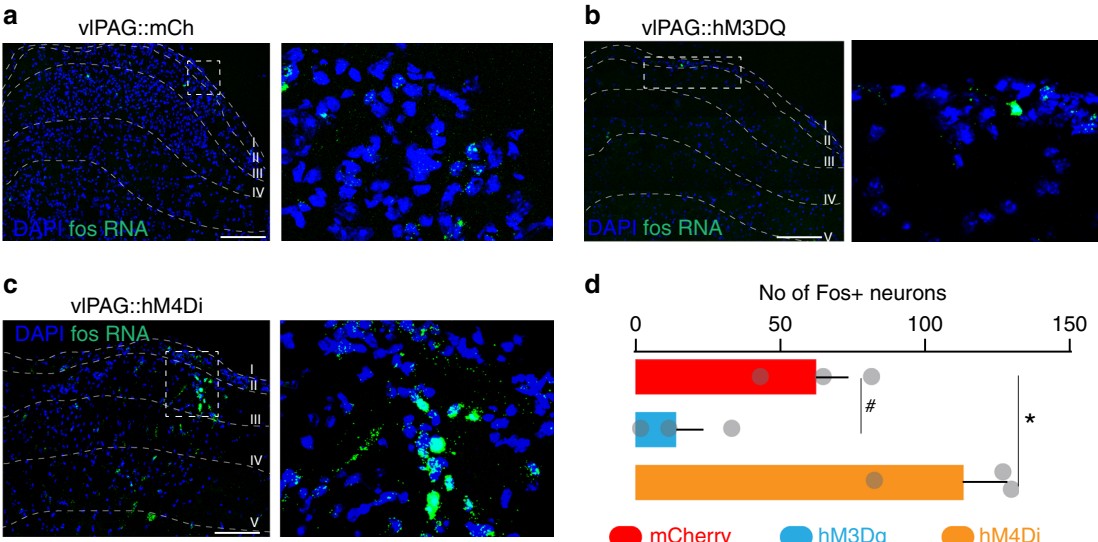

**Fig. 9** vlPAG neurons modulates itch-induced activity in spinal cord. FISH analysis of chloroquine-induced cFos expression in the spinal cord in response to CNO in mice expressing hM3Dq, hM4Di, or mCherry control in vlPAG. **a** Coronal section of spinal cord illustrating c-fos expression following chloroquine treatment in vlPAG::mCh control mice. **b** Chemogenetic activation of PAG neurons decreased expression of c-fos in the spinal cord following chloroquine treatment in vlPAG::hM3Dq mice. **c** Chemogenetic inhibition of PAG neurons increased expression of c-fos in the spinal cord following chloroquine treatment in vlPAG::hM4Di mice. **d** Quantification of spinal chloroquine-induced c-fos expression show decreases in the c-fos staining following CNO treatment in the vlPAG::hM3Dq mice and significant increases in the staining in vlPAG::hM4Di mice compared to mCherry controls. Interaction; *$P = 0.044$. mCh vs. hM4Di, #$P = 0.056$. $N = 3$/group, one-way ANOVA. Scale bars represent 75 μM. All values are mean ± SEM. Source data are available as a Source Data file

It is unclear how distinct neuronal populations integrate and modulate different sensations. We and others have shown that the vlPAG GABAergic and glutamatergic neurons differentially modulate pain neurotransmission[29,30,35–37,47–51]. Itch and pain exhibit a reciprocal relationship at the behavioral level. For example, itch is suppressed by the mild noxious mechanical stimulus of scratching[14,17,18,52]. Given this inverse relationship of itch and pain and these identified differential roles for these neuronal populations in the vlPAG, we investigated the role of GABAergic and glutamatergic vlPAG neurons in the processing of itch. We observed that vlPAG GABAergic and glutamatergic neurons exhibited opposing effects on pruritic information processing. Activation of vlPAG GABAergic neurons resulted in decreased chloroquine-induced itch while their inhibition led to an increase in chloroquine-induced itch. Conversely, activation of vlPAG glutamatergic neurons resulted in a profound increase in chloroquine-induced itch whereas inhibition of these cells resulted in suppression of chloroquine-induced itch. Similar results were observed in a model of chronic itch, where activation of vlPAG GABAergic neurons or inhibition of vlPAG glutamatergic neurons resulted in attenuation of spontaneous scratching behavior produced by chronic dry skin. These results indicate that decreased activity in vlPAG GABAergic and/or increased activity of glutamatergic neurons may underlie central sensitization of itch in the context of chronic itch. It is likely that the vGlut2 + PAG neurons we studied here overlap with the Tac1+ neurons studied by Gao et al.[42]. They found that activating these neurons produced spontaneous scratching, and as such it is possible that Tac1+ neurons are small subset of vGlut2 + PAG neurons. Taken together, we show evidence for the presence of functionally divergent cell types in the vlPAG that modulate itch-evoked behavioral output and the affective component associated with pruritus. Thus, although pain and itch are processed and transmitted by similar neuroanatomical substrates at peripheral, spinal and supraspinal sites, they are discriminated and processed distinctly at the cellular level.

The vlPAG is known to have neuronal subtypes that exhibit different firing patterns in response to noxious stimuli[53,54]. Recently studies have shown that inhibition of vlPAG GABAergic neurons or activation of glutamatergic neurons results in antinociception, whereas activation of vlPAG GABAergic neurons or inhibition of glutamatergic neurons leads to hypersensitivity to painful stimuli[36]. Interestingly, these results together with our current findings demonstrate that vlPAG GABAergic and glutamatergic neurons process both pain and itch signals inversely. The activity profile of GABAergic and glutamatergic neurons was regulated inversely during chloroquine-evoked scratching, as determined here by measuring $Ca^{2+}$ dynamics. The activity of these neuronal populations is consistent with the behavioral results of our chemogenetic experiments in scratching behavior, suggesting that in vivo calcium imaging faithfully recapitulates the activity of GABAergic and glutamatergic vlPAG neurons during pruritic processing. A drawback of fiber photometry is that it captures the averaged activity across a population of neurons, which might lead to missing or masking any heterogeneity in the responses of individual neurons.

We found that vlPAG Vgat+ and Vglut2+ neurons showed opposing responses during chloroquine-evoked scratching in our photometry recordings. Chloroquine evoked scratching inhibits GABAergic neurons and activates glutamatergic neurons. Interestingly, in GABAergic neurons basal activity is elevated just preceding chloroquine evoked scratching, while this phenomenon is not seen during spontaneous scratching bouts. What would be the underlying cause for this? We propose that GABAergic neuronal activity is encoding the sensory and motivational aspects of itch and this activity is suppressed when scratched, possibly contributing to relief of itch. An assumption in this

model is that relief of itch by scratching is responsible for suppression of activity of GABAergic neurons. Interestingly, we observed rapid elevated activity in the Vglut2+ neurons subsequent to scratching. These differences in the modulatory effects of GABAergic and glutamatergic neurons on itch response suggests the presence of distinct microcircuit features. It is possible that inhibition of Vgat+ neurons might disinhibit the Vglut2 +ve neurons subsequent to scratching during chloroquine-evoked itch. The $Ca^{2+}$ dynamics of Vgat and Vglut2 neurons are consistent with a key role of these neurons in regulating itch processing.

The opposing and unique actions of glutamatergic and GABAergic neurons in the modulation of itch identified here emphasize the importance of using cell-type-specific approaches. This also suggests that further dissection of these neuronal subtypes should be undertaken. We have chosen to assess potential unique roles for glutamatergic and GABAergic neurons for the reasons mentioned above, however the PAG comprises multiple neuronal populations expressing several neurotransmitters and neuropeptides[44–46]. Even among the subpopulations studied here, there are certain to be further subclassifications based on anatomy (e.g., projection targets, source of synaptic inputs) or neurochemistry (e.g., neuropeptides or other co-expressed neuromodulators). It is unlikely that all the GABAergic and glutamatergic vlPAG neurons show the same response to pruritic processing. Prior work[37,48–50] suggests that GABAergic neurons in the PAG are mostly interneurons, and we did not find any downstream projections from these neurons in our studies. PAG glutamatergic neurons are output neurons of the PAG. Differences in the modulatory effects of GABAergic and glutamatergic neurons on itch responses suggest the presence of distinct microcircuit features. Though not addressed directly in our study, it is possible that activation of Vgat neurons that encode itch might directly inhibit the Vglut2 neurons to suppress chloroquine evoked ongoing itch. Future studies may discover additional heterogeneity in GABAergic and glutamatergic neuronal subtypes based on their gene expression and their anatomical downstream targets.

There were some notable discrepancies between our behavioral data obtained from pharmacological and chemogenetic manipulation of global PAG activity compared to cell-type specific PAG manipulations (we have included additional discussion points in the supplementary discussion).

Since itch, like pain, is an aversive sensory experience in humans[17,18,52], we hypothesized that pruritogen-evoked itch would produce an aversive state in mice. Here we demonstrate that itch evoked by chloroquine administration resulted in robust conditioned place aversion, in a dose-dependent manner demonstrating that in mice itch is fundamentally an aversive sensation, consistent with recent studies[34]. We also demonstrated that activation of vlPAG neurons abolishes CPA to chloroquine. Furthermore, activation of vlPAG GABAergic neurons prevented place aversion to chloroquine suggesting that vlPAG GABAergic neurons encode the aversive component of itch. Surprisingly, inhibition of vlPAG glutamatergic neurons did not prevent place aversion to chloroquine, though chemogenetic inhibition of these neurons decreased both acute and chronic itch. These results indicate that vlPAG GABAergic neurons, but not glutamatergic neurons, are capable of encoding the aversive component of itch.

The loss of chloroquine-induced aversion upon activation of vlPAG GABAergic neurons could be due to activation of descending inhibition from vlPAG GABAergic neurons, leading to suppression of transmission of chloroquine-evoked activity in itch-specific sensory pathways at the level of the spinal cord. Alternatively, the disruption of pruritogen-induced sensory transmission could occur within the vlPAG itself. Indeed, activity

of these neurons is suppressed in the context of chloroquine-induced scratching, as demonstrated by photometry recordings in the vlPAG. Such an effect could be mediated by activation of GABAergic projections from the vlPAG to the ventral tegmental area (VTA) or to the amygdala neurons that encode aversion[55]. Recent studies have also shown that VTA dopaminergic neurons are involved in the motivational aspect of itch-evoked scratching[56]. Our results suggest that blockade of CPA could be due to a reduction in the negative affect associated with chloroquine-evoked scratching, or to suppression of transmission of sensory information in itch-specific sensory pathways. Interestingly, recent work using chemogenetic inhibition of vPAG GABAergic neurons suggested that these neurons encode for aversive anxious behaviors[57].

Together, our data demonstrate that chloroquine-induced itch is a fundamentally aversive sensation and can be suppressed by activating vlPAG GABAergic neurons. We conclude that the vlPAG is a hub that can bidirectionally modulate itch processing in the context of acute and chronic itch. This differential modulation of itch by glutamatergic and GABAergic vlPAG neurons is reciprocal in nature to the role of these same cell populations in the modulation of pain[36] (Fig. 10). Our results showing that activation of vlPAG GABAergic neurons or inhibition of vlPAG glutamatergic neurons can attenuate chronic itch suggesting that disengagement in this circuit has the potential to contribute to exacerbation of chronic itch.

The vlPAG is well-positioned to be modulated by top-down control from neural regions important for processing pain and itch[1,9,17,18,24,29,52,58], including the somatosensory, anterior cingulate, and insular cortices, as well as the thalamus[59,60]. The vlPAG has been shown to be instrumental in descending modulation of pain processing[20,37]. Our current observations identify a previously unknown role of the vlPAG in modulation of itch

wherein selective activation and inhibition of specific subsets of vlPAG GABAergic or glutamatergic neurons produces distinct phenotypes. These findings point to the essential role of descending supraspinal circuitry in modulating itch processing, and also support previous studies suggesting the likely existence of strong descending itch modulation pathways[14,15]. Finally, the unexpected differences observed between global pharmacologic inhibition, global modulation of resident neuronal activity and cell-type-selective modulation illustrate the importance of cell-type-specific approaches, and provide a clear case for their use in reexamination of basic tenets of neural circuitry.

## Methods

**Animals**. All experiments were conducted in accordance with the National Institute of Health guidelines and with approval from the Animal Care and Use Committee of Washington University School of Medicine. Mice were housed on a 12-h light-dark cycle (6:00 a.m. to 6:00 p.m.) and were allowed free access to food and water. All animals were bred onto C57BL/6J background and no more than 5 animals were housed per cage. Male littermates between 8–12 weeks old were used for experiments. Slc32a1tm2Lowl (Vgat-ires-Cre), Slc17a6tm2Lowl (Vglut2-ires-Cre), and C57BL/6J mice were purchased from Jackson Laboratories and colonies were established in our facilities. Litters and animals were randomized at the time of assigning experimental conditions for the whole study. Experimenters were blind to treatment and genotype.

**Viral constructs**. Purified and concentrated adeno-associated viruses coding for Cre-independent hM3Dq-mCherry (rAAV8/hSyn-hM3Dq-mCherry; $3.2 \times 10^{12}$ particles/ml, Lot number: AV5359 and Lot date: 6/13/2013), hM4Di-mCherry (rAAV8/hSyn-hM4Di-mCherry; $2 \times 10^{12}$ particles/ml, Lot number: AV5360 and Lot date: 6/6/2013),and hSyn-eGFP (rAAV8/hSyn-eGFP; $8 \times 10^{12}$ particles/ml, Lot number: AV5075 and Lot date: 10/12/2012) was used as used as control for non Cre dependent studies in C57 mice. Cre-dependent hM3Dq-mCherry(rAAV5/hSyn-DIO-hM3Dq-mCherry; $6 \times 10^{12}$ particles/ml, Lot number: AV4495c and Lot date: 02/23/2012) and hM4D-mCherry (rAAV5/hSyn-DIO-hM4Di-mCherry; $6 \times 10^{12}$ particles/ml, Lot number: AV4496c and Lot date: 11/20/2012) Control mCherry (rAAV5/hSyn-DIO-mCherry; $3.4 \times 10^{12}$ particles/ml, Lot number: AV5360 and Lot date: 04/09/2015) was used to express in the Vgat and Vglut2 Cre mice. All vectors were packaged by the University of North Carolina Vector Core Facility. All vectors were aliquoted and stored in −80 °C until use. We functionally verified and validated DREADD constructs we used in this study in our prior publication[36].

**Stereotaxic Surgeries**. Mice were anesthetized with 1.5 to 2.0% isoflurane in a gas chamber using isoflurane/breathing air mix. Once deeply anesthetized, they were placed and secured in a stereotactic frame (David Kopf Instruments, Tujunga, CA) where surgical anesthesia was maintained using 2% isoflurane. Mice were kept on a heating pad for the duration of the procedure. Preoperative care included application of sterile eye ointment for lubrication, administration of 1 mL of sub-cutaneous saline and surgery site sterilization with iodine solution. A small midline dorsal incision was performed to expose the skull and viral injections were performed using the following coordinates: vlPAG, −4.8 to 4.9 mm from bregma, +/− 0.3 to 0.4 mm lateral from midline and 2.7 to 2.9 mm ventral to skull. Viruses were delivered bilaterally, using a stereotaxic mounted syringe pump (Micro4 Micro-syringe Pump Controller from World Precision Instruments) and a custom 2.0 µL Hamilton syringe. Injections of 150 nL of the desired viral vectors into the area of interest were performed at a rate of 1 µL per 10 min. We allowed for a 10-min period post injection for bolus diffusion before removing the injection needle. Postoperative care included closure of the cranial incision with sutures and using veterinary tissue adhesive and application of topical triple antibiotic near the incision site. Animals were monitored while on a heating pad until they full recovery from the anesthetic.

**Cannula Implantation and lidocaine infusion**. The surgical protocol was the same as the described above for viral injections. Stainless steel guide cannulas (8 mm) were implanted at the vlPAG and fixed to the skull using two bone screws (CMA anchor screws, #7431021) and dental cement. An 8 mm stylet was used to avoid guide cannula clogging. Mice were allowed to recover for 7 days. Microinjections of 0.25 µl 4% lidocaine (Sigma, L1663) diluted in 0.9% saline were performed 15 min before pruritic agent induced scratching behavior was assessed. Mice were later euthanized, and cannula placements were verified using cresyl violet nissl stain. Animals in which cannula's were not placed in the vlPAG are excluded from the study.

**Chemogenetic manipulation**. For non-specific chemogenetic control of vlPAG neurons, C57BL/6J mice were injected with non-Cre dependent control eGFP, hM3Dq or hM4Di viruses. For subpopulation specific chemogenetic control of

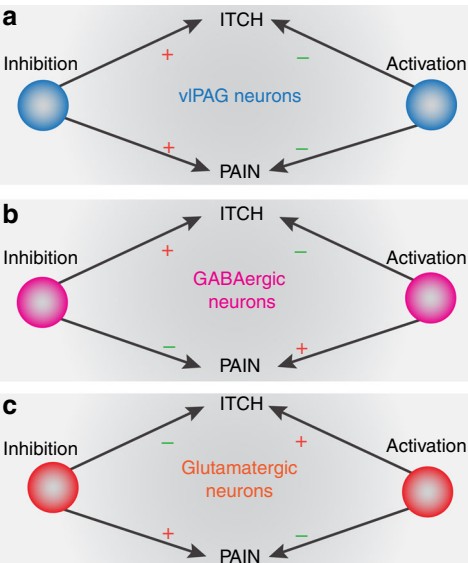

**Fig. 10** Summary of the effects of manipulation of global vs. cell type-specific activity in the vlPAG on pain and itch. **a** Global (non-cell-type-specific) chemogenetic activation of vlPAG neurons resulted in robust suppression of itch and pain, while global chemogenetic inhibition of vlPAG neurons resulted in profound increases in itch and pain behaviors.
**b** Activation of vlPAG GABAergic neurons attenuates itch and enhances pain while inhibition of vlPAG GABAergic neurons leads to increased itch and decreased pain. **c** Activation of vlPAG glutamatergic neurons leads to enhanced itch and decreased pain behaviors, while inhibition of vlPAG glutamatergic neurons attenuates itch and potentiates pain

vlPAG neurons Vgat-ires-Cre and Vglut2-ires-Cre mice were injected with Cre dependent control mCh, hm3Di or hm4Dq viruses. DREADD constructs used in this study were validated previously for their functional expression in the PAG, including their ability to increase (hM3Dq) or decrease (hM4Di) neuronal firing in slices from animals expressing these viral constructs[36]. Three weeks later mice were injected with (C57BL/6 J, VGlut2 Cre and VGAT Cre, respectively) clozapine N-oxide (CNO, BML-NS105 from Enzo life sciences) 2 h before doing behavioral experiments and data were collected between the second-hour and third-hour post-injection. All baselines for pruritic responses were recorded 2 weeks after the virus injections and one week prior to the CNO administration. We performed dose response studies of CNO to determine the optimal dose for use in behavioral experiments. Based on these dose-finding studies, we selected 5 mg/kg CNO as a dose of CNO that produces near maximal effects in Cre dependent DREADD-expressing animals and showed no signs of behavioral affects in control vector-expressing animals. For non-Cre dependent studies, we observed that a dose above 1 mg/kg CNO was lethal, so we used 1 mg/kg CNO dose for both activation and inhibition of vlPAG neurons.

**Pruritic agent induced scratching behaviors.** The nape of the neck mice was shaved one day prior to experiments. Mice then were placed in clear plexiglass behavioral boxes for at least two hours for acclimation for baselines[41]. For chemogenetic manipulations, CNO was administered before placing the mice in the plexiglass behavioral boxes and chloroquine (200 µg/50 µl, nape of the neck) induced scratching behavior was performed 90 min after the CNO administration.

**Acetone and ether, water (AEW) induced itch.** To model dry skin-mediated itch, spontaneous scratching behavior was induced by treatment twice daily with an acetone/diethylether (1:1) solution for 15 s followed by water for 30 s on the nape of the neck. Following seven days of AEW treatment, spontaneous behavior was video recorded for two hour pre-CNO and post-CNO. Scratch bouts directed to the nape of the neck and rostral back were counted in 5-min intervals during blinded off-line analysis.

**Conditioned place aversion (CPA).** CPA was performed using an unbiased, counterbalanced three-compartment conditioning apparatus. Each chamber had a unique combination of visual, properties (one side had black and white vertical walls, whereas the other side had black and white striped walls). On the pre-conditioning day (day 1), mice were allowed free access to all three chambers for 20 min. Behavioral activity in each compartment was monitored and recorded with a video camera and analyzed using Ethovision 8.5 (Noldus) or ANY-Maze soft-ward. Mice were randomly assigned to saline and chloroquine compartments and received a saline injection (50 µl) on the nape of the neck and on the mouse caudal back, in the morning and a chloroquine injection (400 µg/50 µl) on the nape of the neck and on the mouse caudal back in the afternoon, at least 4 h after the morning training on 3 consecutive days (Day 2, 3, and 4). To enhance the association of chloroquine induced scratching behavior with the paired chamber, we administered chloroquine and left the mice in their holding cage for 4 min, then placed them in the paired chamber during the time of the peak scratching response (20 min in the chamber). To assess for place aversion, the mice were then allowed free access to all three compartments on day 5 for 30 min[61]. Scores were calculated by subtracting the time spent in the chloroquine-paired compartment, post-test minus the pre-test. To test the effect of DREADD hM3Dq activation on chloroquine-induced place aversion, mice injected with AAV8 hM3Dq–mCherry and AAV8 EGFP were allowed free access to all three chambers for 30 min on the pre-conditioning day (day 1). On day 2, 3, and 4, both mice cohorts received a saline injection (50 µl) on the nape of the neck and on the mouse caudal back, and this chamber is paired with systemic saline injection 2 h before they were placed in the compartment in the morning and a chloroquine injection (400 µg/50 µl) on the nape of the neck and on the mouse caudal back and this chamber is paired with systemic CNO injection 2 h before they were placed in the compartment in the afternoon. To test the effect of DREADD hM3Dq activation on chloroquine-induced place aversion, the mice were allowed free access to the three compart-ments on day 5 for 30 min. Scores were calculated by subtracting the time spent in the chloroquine-paired compartment, post-test minus the pre-test.

**Fiber Photometry.** For in vivo calcium imaging of vlPAG GABAergic and gluta-matergic neurons, we injected vlPAG with Cre-dependent GCaMP6s (AAV-DJ EF1a-DIO-GCaMP6s, $3 \times 10^{13}$ particles/ml, Stanford vector core). Fiber optic probes were unilaterally implanted above the left vlPAG (−4.8 to 4.9 mm from bregma, +/− 0.3 to 0.4 mm lateral from midline and 2.7 to 2.9 mm ventral to skull). After 4 weeks of viral expression, mice were handled and acclimated, tethered as during imaging, for 7 days in the test behavioral chamber. On test day, mice were habituated with the tethered fiber optic patch cord (0.48NA, BFH48-400, Doric lenses) in the test chamber (15 × 15 cm) for 30 min and injected with chloroquine (200ug/50ul) on the nape of the neck and recordings were performed.

A fiber optic patch cord was used to connect to the fiber implant and deliver light to excite and record the GCAMP signal using a custom-built fiber photometry rig. The fiber photometry recording setup was built with some modifications to previously described specifications[38]. Fluorescence excitation was provided by two

LEDs at 211 and 537 Hz to avoid picking up room lighting (M405FP1, M470F1; LED driver: LEDD1B; Thorlabs) was bandpass filtered (FMC1 + (405/10) -(475/28)_(525/45)_FC), Doric lenses) and delivered to the vlPAG to excite GCaMP6s. The emitted light was bandpass filtered (FMC1 + _(405/10)-(475/28)_(525/45)_FC), Doric lenses) and sent to a photoreceiver to detect the signal (Newport, 2151). The signal from the photoreceiver was recorded using a RZ5P real-time processor (TDT). Data were acquired at 10 kHz and then demodulated at 211 and 537 Hz. The demodulated signal was then low-pass filtered (4 Hz) in a custom MATLAB script. The extracted 405 nm signal was then scaled to fit the GCaMP signal for the recording session. To isolate the movement-corrected GCaMP signal from channel, we subtracted the signal at 405 nm from the 475 nm GCaMP signal. dF/F was obtained by dividing the final signal with its mean value. Behavioral event timestamps associated with chloroquine evoked scratching behavior were scored and aligned with GCAMP signal in the MATLAB script to create pre-stimulus and peri-stimulus time bins. To obtain pre-stimulus and peri-stimulus chloroquine evoked scratching events, if the scratching events happened to close to each other in a 30 s window, they were combined and called as one event. Z-score was obtained by subtracting the mean of the GCaMP signal from the bin value of the GCaMP signal and dividing it with the standard deviation of the bin value of the GCaMP signal.

**Fluorescence in situ hybridization (FISH).** For non-specific chemogenetic control of vlPAG neurons, C57BL/6 J mice were injected with non-Cre dependent control mCh, hM3Dq or hM4Di viruses. Three weeks later mice were injected with CNO (BML-NS105 from Enzo life sciences) 2 hrs before chloroquine was injected on the nape of the neck. Thirty minutes post chloroquine administration, mice were rapidly decapitated, cervical spinal cords were dissected and flash frozen in −50 °C 2-methylbutane and stored at −80 °C for further processing. Coronal sections of the spinal cord corresponding to the cervical level, were cut at 15 µM at −20°C and thaw-mounted onto Super Frost Plus slides (Fisher). Slides were stored at −80 °C until further processing. FISH was performed according to the RNAScope® 2.0 Fluorescent Multiple Kit v2 User Manual for Fresh Frozen Tissue (Advanced Cell Diagnostics, Inc.)[62]. Slides containing spinal cord sections were fixed in 4% para-formaldehyde, dehydrated, and pretreated with protease IV solution for 30 min. Sections were then incubated with target probes for mouse cFos (mm-Fos, catalog number 316921, Advanced Cell Diagnostics). Following probe hybridization, sec-tions underwent a series of probe signal amplification steps (AMP1–4) followed by incubation of fluorescent probes (Alexa 488, Opal 470), designed to target the specified channel associated with the probes. Slides were counterstained with DAPI and coverslips were mounted with Vectashield Hard Set mounting medium (Vector Laboratories). Images were obtained on a Leica TCS SPE confocal microscope (Leica), and Application Suite Advanced Fluorescence (LAS AF) software was used for analyses. To quantify number of cFos[+ve] cells, we counted DAPI-stained nuclei that coexpress minimum of five cFos puncta as an cFos[+ve] cell. We did not include any cFos puncta that does not overlay on top of the DAPI-stained nuclei as part of our analysis.

**Immunohistochemistry.** Adult mice were deeply anesthetized using a ketamine/xylazine cocktail and then perfused with 20 ml of with phosphate-buffered saline (PBS) and 4% paraformaldehyde (weight/volume) in PBS (PFA; 4 °C). Brains were carefully removed, post fixed in 4% PFA overnight and later immersed in 30% sucrose for at least 48 h. Tissues were mounted in OCT while allowing solidifica-tion of the mounting medium at −80 °C. Using a cryostat, 30µm tissue sections were collected and stored in PBS1× 0.4% sodium azide at 4 °C. After washing the sections in PBS1×, we blocked using 5% normal goat serum and 0.2% Triton-X PBS 1× for one hour at room temperature. Primary antibodies against mCherry (Mouse, Clontech, 632543; 1/500) and GFP (rabbit polyclonal anti-GFP, Aves A11122; 1/500) were diluted in blocking solution and incubated overnight at 4 °C. After three 10-min washes in PBS1×, tissues were incubated for one hour at room temperature with secondary antibodies (Life Technologies: Alexa Fluor488 donkey anti rabbit IgG (1/300); Alexa Fluor 488 goat anti rabbit (1/300); Alexa Fluor 555 goat anti mouse (1/300)) and Neurotrace (435/455 nm, 1/500) at room tempera-ture. Three PBS1× washes followed before sections were mounted with Vectashield (H-1400) hard mounting media and imaged after slides cured. Images were obtained on a Nikon Eclipse 80i epifluorescence microscope[63–66].

**Statistics.** Throughout the study, researchers were blinded to all experimental conditions. Exclusion criteria for our study consisted on a failure to localize expression in our experimental models or off-site administration of virus or drug. At least 3 replicates measurements were performed and averaged in all behavioral assays. The number of animals used is indicated by the "N" in each experiment. When paired $t$ test was used for comparing paired observations, we evaluated for normality using the D'Agostino and Pearson omnibus normality test for all datasets. Therefore, only when normality could be assumed, we used a parametric test to analyze out data. If normality could not be assumed, a nonparametric test or a Wilcoxon matched pairs text was used to evaluate differences between the means of our experimental groups. Two-way ANOVA was used for comparing between different control and treatment groups. Bonferroni's post hoc tests were used

(when significant main effects were found) to compare effects of variables. A value of $p < 0.05$ was considered statistically significant for all statistical comparisons.

## Data availability
The data that support the findings of this study are available from the corresponding author upon reasonable request. Source data underlying Figs. 1–9 and Supplementary Figs. 1–3 are available as a Source Data file.

## Code availability
Matlab code used in this study is available from the corresponding author upon reasonable request.

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

## Acknowledgements

This work was funded by NINDS R01NS106953 to RWG, Urology Care Foundation Research Scholars Program and Kailash Kedia Research Scholar award and NIDDK Career development award (K01 DK115634) to VKS, and the Medical Scientist Training Program (MSTP) Grant T32GM07200 and NINDS NRSA 5F31NS103472-02 to JGGR. We thank Christian Pedersen and Michael R Bruchas for their help with Matlab code for the photometry analysis. We thank Sherri Vogt for her assistance with mouse colony maintenance and genotyping. We would like to thank Dr. Steve Davidson, Dr. Judy Golden and Dr. Bryan Copits for helpful discussion with the manuscript and experimental design; we would like to thank all the Gereau lab members for their help with manuscript preparation.

## Author contributions

V.K.S and R.W.G. designed the experiments; V.K.S., J.G.G., S.S. performed anatomical analyses, and behavior; V.K.S and J.J.Y. performed FISH experiments; V.K.S., J.G.G., and R.W.G. analyzed the data; V.K.S., and R.W.G. wrote the manuscript with comments from all the authors.

## Additional information

**Competing interests:** The authors declare no competing interests.

