## [Peer Review File · Nature Communications]

Reviewers' comments:

Reviewer #1 (Remarks to the Author):

Neural circuits of itch sensation have been intensively studied recently. The current study employs a multidisciplinary approach including mouse genetics, chemo-genetics, in vivo imaging, and behavioral analysis to dissect central nervous system pathways regulating itch transmission. The authors identified two specific groups of neurons in the periaqueductal gray (PAG) (i.e. vGlut2+ and Vgat+ neurons) play opposite effects on itch response. By specifically activating and inhibiting these neurons, they found that glutamatergic vGlut2 neurons facilitate itch whereas GABAergic Vgat neurons suppress itch. Interestingly, Vgat neurons and not vGlut2 neurons mediate the aversive aspect of itch. Together, these data strongly suggest PAG in the brain is a major site for regulating itch sensation.

The study has nicely demonstrated the important role of PAG in itch regulation. In fact, the current finding is supported by a recent publication in *Neuron* (Gao et al., 2018). These two studies are nicely reinforced each other. However, the current study goes further by characterizing the inhibitory role of Vgat. There are several minor concerns:

- 1) Are vGlut2+ neurons are the same Tac1+ neurons studied by Gao et al? If not, it will suggest there are multiple subsets of PAG neurons can facilitate itch.
- 2) Why does global activation of PAG neurons has a net negative impact on itch response but the two subsets (vGlut2 and Vgat) have opposite effects? Does it mean the two subsets function epistatically rather in parallel? The authors should provide their thoughts on this.
- 3) In Figure 3d, there is a noticeable increase in Vgat+ neuron activities before the itch started (time 0) and then the activities went down to baseline. Is there an explanation for this increase before seeing decrease?

Reviewer #2 (Remarks to the Author):

Authors demonstrate that PAG contributes to itch modulation in behavioral mice. In general, the study is straightforward, and lacks of novel approaches and new signaling mechanisms. Behavioral results are different to distinguish neuronal mechanism for how PAG may affect itch-like responses. Direct evidence are needed to show if itching neuronal transmission is affected at the spinal cord dorsal horn at least. Furthermore, PAG may indirectly affect brain responses to itching, it may be possible that such manipulation affect itching related process at other brain areas such as amygdala as authors mentioned. If they can reveal such connection, it may improve the impact of the paper. The results from excitatory vs GABA neurons are not completely novel, and are quite predictive. The study also has less impact for translational aspect of treating chronic itching. In sum, the paper lacks of novel mechanism and the method is not also innovative. I recommend that this paper is more appropriate for professional journal such as *Pain* or *Journal of Pain*.

We thank the reviewers for their enthusiasm, and the helpful comments and suggestions. We have carefully considered all of these points. Our responses and the changes we have made to the manuscript are detailed below. Reviewer's full comments are shown in blue text, and our responses are shown in italics. As a result of these changes, we think this manuscript is much improved, and we hope that it is now suitable for publication in *Nature Communications*.

Reviewer 1:

Neural circuits of itch sensation have been intensively studied recently. The current study employs a multidisciplinary approach including mouse genetics, chemo-genetics, in vivo imaging, and behavioral analysis to dissect central nervous system pathways regulating itch transmission. The authors identified two specific groups of neurons in the periaqueductal gray (PAG) (i.e. vGlut2+ and Vgat+ neurons) play opposite effects on itch response. By specifically activating and inhibiting these neurons, they found that glutamatergic vGlut2 neurons facilitate itch whereas GABAergic Vgat neurons suppress itch. Interestingly, Vgat neurons and not vGlut2 neurons mediate the aversive aspect of itch. Together, these data strongly suggest PAG in the brain is a major site for regulating itch sensation. The study has nicely demonstrated the important role of PAG in itch regulation. In fact, the current finding is supported by a recent publication in *Neuron* (Gao et al., 2018). These two studies are nicely reinforced each other. However, the current study goes further by characterizing the inhibitory role of Vgat. There are several minor concerns:

1) Are vGlut2+ neurons are the same Tac1+ neurons studied by Gao et al? If not, it will suggest there are multiple subsets of PAG neurons can facilitate itch.

Our response: *It is likely that the vGlut2+ PAG neurons we studied here overlap with the Tac1+ neurons studied by Gao et al. Gao et al found that activating these neurons produced spontaneous scratching, and as such it is possible that Tac1+ neurons are small subset of vGlut2+ PAG neurons. This is now discussed on page 12, line 18 to 21 of the revised manuscript.*

2) Why does global activation of PAG neurons has a net negative impact on itch response but the two subsets (vGlut2 and Vgat) have opposite effects? Does it mean the two subsets function epistatically rather in parallel? The authors should provide their thoughts on this.

Our response: *This is very interesting question. Based on our results it seems like Vgat+ neurons play a predominant role in the net impact of the PAG on itch. It is also possible the viral transfection*

efficiency between cell-types might also play a role in the final outcome of the behavioral output as we are injecting very small volumes of virus.

Prior work suggests that GABAergic neurons in the PAG are mostly interneurons, and we did not find any downstream projections from these neurons in our studies. PAG glutamatergic neurons are output neurons of the PAG. Differences in the modulatory effects of GABAergic and glutamatergic neurons on itch response suggest the presence of distinct microcircuit features. Though not addressed directly in our study, it is possible that activation of Vgat+ neurons that encode itch might directly inhibit the Vglut2+ve neurons to suppress chloroquine evoked ongoing itch. This possibility is now discussed on page 13, line 30 to page 14, line 3 of the revised manuscript.

3) In Figure 3d, there is a noticeable increase in Vgat+ neuron activities before the itch started (time 0) and then the activities went down to baseline. Is there an explanation for this increase before seeing decrease?

Our response: We hypothesize that this ramping up in activity in these neurons preceding a scratch bout in chloroquine treated mice might be encoding the intensity of itch as we don't see this phenomenon in mice that spontaneously scratch, based on some additional analysis we have now performed, where we found no significant differences in the Vgat Ca²⁺ dynamics between baseline and post scratching bout for spontaneous scratching events (Figure S3). There is a significant difference in the change in Vgat Ca²⁺ activity when compared between spontaneous scratching at baseline (pre-chloroquine) and that seen associated with chloroquine-evoked scratching bouts.

We have now included this additional data set in supplementary figure 3, and the following text was added to page 7 line 24 to line 27 of the revised manuscript:

“We did not observe similar robust changes in Ca²⁺ dynamics during spontaneous scratching bouts in the Vgat+ neurons prior to chloroquine evoked scratching (Figure S3). The decrease in the Vgat Ca²⁺ activity was significantly greater during chloroquine-evoked scratching events compared to these spontaneous scratches recorded prior to chloroquine injection.”

We have also made following changes in the discussion in page 13 line 7 to line 20.

“We found that vIPAG Vgat+ and Vglut2+ neurons showed opposing responses during chloroquine-evoked scratching in our photometry recordings. Chloroquine evoked scratching inhibits GABAergic neurons and activates glutamatergic neurons. Interestingly, in GABAergic neurons basal activity is elevated just preceding chloroquine evoked scratching, while this phenomenon is not seen during spontaneous scratching bouts. What would be the underlying cause for this? We propose that GABAergic neuronal activity is encoding the sensory and motivational aspects of itch and this activity is suppressed when scratched, possibly contributing to relief of itch. An assumption in this model is that relief of itch by scratching is responsible for suppression of activity of GABAergic neurons. Interestingly, we observed rapid elevated activity in the Vglut2+ neurons subsequent to scratching. These differences in the modulatory effects of GABAergic and glutamatergic neurons on itch response suggests the presence of distinct microcircuit features. It is possible that inhibition of Vgat+ neurons might disinhibit the Vglut2+ve neurons subsequent to scratching during chloroquine-evoked itch. The Ca²⁺ dynamics of Vgat and Vglut2 neurons are consistent with a key role of these neurons in regulating itch processing.”

Reviewer 2:

Reviewer 2 comments on lack of novelty: Authors demonstrate that PAG contributes to itch modulation in behavioral mice. In general, the study is straightforward, and lacks of novel approaches and new signaling mechanisms.... (and later) ... The results from excitatory vs GABA neurons are not completely novel, and are quite predictive. The study also has less impact for translational aspect of treating chronic itching. In sum, the paper lacks of novel mechanism and the method is not also innovative. I recommend that this paper is more appropriate for professional journal such as Pain or Journal of Pain.

Our response: *We are disappointed to hear the reviewer’s general lack of enthusiasm regarding our manuscript. We strongly disagree with the reviewer’s suggestion that this manuscript lacks*

novelty. We would like to highlight the key findings of our manuscript that adds substantial knowledge to supra spinal pathways regulating sensory and affective component of itch transmission.

Here we used a combination of pharmacologic, chemogenetic and optical imaging approaches to demonstrate that the vIPAG contains circuitry that robustly modulates sensory and affective components of itch, identifying this region as a key neuromodulatory hub for aversive stimuli, including pain and itch. The most significant findings, however, were achieved using cell-type specific monitoring of real-time Ca^{2+} dynamics and chemogenetic modulation to carefully dissect the role of PAG GABAergic and glutamatergic neurons in the sensory vs. affective components of itch. This type of analysis has not been reported for PAG neurons in the modulation of pruritic processing. Using these approaches, we made the surprising finding that activity of GABAergic and glutamatergic neurons in the PAG is modulated in an opposing manner during pruritogen-evoked scratching. Furthermore, activation of GABAergic neurons or inhibition of glutamatergic neurons in the vIPAG resulted in attenuation of chloroquine-evoked scratching and spontaneous scratching behavior in a model of chronic dry skin-induced pruritis.

A second major finding reported here is related to studies examining the affective component of itch. We adapted the conditioned place aversion assay to unequivocally demonstrate the aversive nature of itch and utilized chemogenetic manipulation to demonstrate that the vIPAG can regulate this aversion. Remarkably, we observed that PAG GABAergic neurons, but not glutamatergic neurons, encode the aversive component of itch. Thus, the PAG represents a neuromodulatory hub that regulates acute and chronic itch, as well as itch-related aversion.

To our knowledge, these are the first studies to demonstrate modulation of the aversive component of itch by supraspinal central circuits using CPA to chloroquine. Furthermore, our results also suggest that the PAG is poised to tightly regulate spontaneous scratching observed in chronic itch conditions. Thus, the present studies provide a blueprint for future investigations studying the neurobiology of sensory and affective component of itch.

Reviewer 2 comment 2: Behavioral results are different to distinguish neuronal mechanism for how PAG may affect itch-like responses. Direct evidence are needed to show if itching neuronal transmission is affected at the spinal cord dorsal horn at least.

Our response: This was an excellent suggestion – to test whether spinal cord neuronal transmission is affected by manipulating PAG neurons. Our laboratory is not equipped to directly measure spinal cord neuronal activity via *in vivo* electrophysiology during scratching behaviors. Fortunately, we were able to address this request from the reviewer by performing additional studies wherein we used activity-dependent mapping studies using cFos labelling (a surrogate marker for neuronal activity) to study whether spinal cord activity is altered with PAG DREADD manipulations. Using fluorescent *in situ* hybridization (RNAscope) for Fos, we found that global chemogenetic activation of PAG neurons suppressed spinal cFos induced by chloroquine (see revised Figure 6 b & d), whereas global chemogenetic inhibition of PAG neurons lead to increases

in the spinal cFos to chloroquine (Figure 6 c & d). These results suggest that PAG neurons modulate spinal pruritic processing possibly by descending projections to spinal cord via rostral ventromedial medulla (RVM) as suggested by Gao et al. 2019. This finding adds substantially to the paper, and we thank the reviewer for the recommendation.

We have included these additional data in the revised manuscript as a new Figure (Fig 6 – see above). The following text was also added to page 11 line 4 to 12 of the revised manuscript:

“To determine whether global manipulation of PAG neurons modulates pruritic processing via descending control at the level of the spinal cord, we performed activity-dependent mapping studies using cFos labelling (a surrogate marker for neuronal activity). We found chloroquine injection induced significant cFos expression in the spinal cord, and that global chemogenetic activation of PAG neurons suppressed this chloroquine-induced induction of spinal cFos

expression (Figure 6 b & d), whereas global chemogenetic inhibition of PAG neurons lead to increases in the spinal cFos to chloroquine stimuli (Figure 6 c & d). These results suggest that PAG neurons may modulate spinal pruritic processing by descending projections to the spinal cord via the rostral ventromedial medulla (RVM), as suggested by recent work⁴²”

The following text was also added to the discussion on page 15 line 7 to line 12:

“We found that manipulating the activity of PAG neurons during chloroquine-evoked scratching leads to alterations in spinal cord activity, as we assessed using cFos expression studies, suggesting that vIPAG neurons modulate spinal pruritic processing. The PAG forms strong connections with the amygdala, the habenula, several thalamic and hypothalamic nuclei, the RVM and the locus coeruleus^{24,47,51,55-59}. We hypothesize based on prior work⁴² that PAG neurons might be modulating spinal cord activity via projections to RVM.”

Reviewer 2 comment 3: Furthermore, PAG may indirectly affect brain responses to itching, it may be possible that such manipulation affect itching related process at other brain areas such as amygdala as authors mentioned. If they can reveal such connection, it may improve the impact of the paper.

Our response: We agree with this reviewer’s excellent point. We have complementary data to this manuscript suggesting recruitment of the amygdala in pruritic modulation. Currently a large manuscript including these data is under consideration elsewhere. As that work is currently under consideration, we do not think it is appropriate to include the data here.

Reviewer 3:

This was an interesting manuscript with several notable strengths. The authors used converging approaches to explore the role of vIPAG neuronal subtypes in itch. Correlations between number of transfected cells and behavior are remarkable and make a very strong case.

Our response: We appreciate reviewer’s enthusiasm for work presented here.

There are a few issues:

1) Behavioral data for intra-vIPAG lidocaine infusion, pain-vIPAG chemogenetics, and cell-type specific vIPAG chemogenetics has discrepancies - lidocaine [so pan-inhibition of all neurons and terminals in the region] reduces scratching, pan-vIPAG activation reduces scratching and

inhibition increases scratching, and vIPAG GABA activation reduces scratching whereas vIPAG Glu increases scratching. This suggests that vIPAG GABA is either a more abundant cell type or a functionally stronger cell type than vIPAG Glu [and also that vIPAG Glu or its disinhibition via terminals is more susceptible to lidocaine?], but this is not discussed or tested.

Our response: *We thank the reviewer for pointing out that this topic needed to be discussed more clearly. To address this, we have included our views on these discrepancies in the discussion, clarifying the significance of the various findings and the importance of the use of cell type-specific approaches, in the revised manuscript. These changes can be found in page 14 line 6 to line 24 of the revised manuscript, and are included below:*

“There were some notable discrepancies between our behavioral data obtained from pharmacological and chemogenetic manipulation of global PAG activity compared to cell-type specific PAG manipulations. First, global vIPAG inactivation by lidocaine infusion attenuated scratching behaviors, an effect that was not mimicked by global chemogenetic inhibition of vIPAG activity. In fact, global chemogenetic inhibition in the vIPAG enhanced scratching behavior, whereas chemogenetic activation of vIPAG suppressed scratching. This distinction can be explained by either a potential impact of lidocaine on axons of passage in the PAG, or by a predominant or epistatic effect of lidocaine on a specific cell type. Consistent with the latter concept, cell-type-specific chemogenetic inhibition of vGat and Vglut2 neurons had opposing actions, where inhibition of Vglut neurons caused suppression of scratching and inhibition of Vgat neurons enhanced scratching behaviors. The overall effect of lidocaine therefore could be explained by a prominent role of inhibition of the Vglut2 neurons. Furthermore, activating Vgat neurons attenuated scratching, mimicking the effects of global PAG activation. It is tempting to speculate that these results can be explained by a microcircuit in the vIPAG, where Vgat neurons exert inhibitory control over Vglut neurons, which in turn provide descending projections that can enhance itch behaviors. These results clearly highlight the differences in pharmacological, global and cell-type specific manipulations, and as such the behavioral outcomes using global pharmacological, electrophysiological, or other types of stimulation or inhibition need to be carefully interpreted.”

2) Issues with fiber photometry: Generally needs to be better at distinguishing scratching behavior and itch treatment states. So it's good to know that there is a difference between baseline and a scratching bout, but an equally important/relevant question here is whether a scratching bout during baseline and a scratching bout during chloroquine treatment look the same, and whether

activity between BL [no itch] vs chloroquine [itch] differs as well. The Vgat fiber photometry data isn't convincing, especially that sample trace [it's not reducing activity like they claim, it's returning to baseline]. I would recommend doing a peristimulus histogram to look at scratching bouts during BL and chloroquine treatment (if feasible), but also a quantification of BL vs chloroquine treatment in general.

Our response: We appreciate reviewer's concerns. We have done additional analysis and scored calcium dynamics in Vgat PAG neurons during spontaneous scratching events. We did not observe similar robust changes in Ca^{2+} dynamics during spontaneous scratching bouts in the Vgat+ neurons prior to chloroquine evoked scratching (Figure S3). The decrease in the Vgat Ca^{2+} activity was significantly greater during chloroquine-evoked scratching events compared to these spontaneous scratches recorded prior to chloroquine injection.

Our Modifications to the manuscript: We have now included this additional data set in Supplementary Figure 3, and the following text was added to page 7 line 24 to line 27 of the revised manuscript:

"We did not observe similar robust changes in Ca^{2+} dynamics during spontaneous scratching bouts in the Vgat+ neurons prior to chloroquine evoked scratching (Figure S3). The decrease in the Vgat Ca^{2+} activity was significantly greater during chloroquine-evoked scratching events compared to these spontaneous scratches recorded prior to chloroquine injection

And further, to amplify on this sub-comment from above: The Vgat fiber photometry data isn't convincing, especially that sample trace [it's not reducing activity like they claim, it's returning to baseline].

We thank the reviewer for raising this point. In response to this comment, we have re-run our analysis to include a larger time window prior the scratching bouts, and have updated the sample trace figures to include the additional duration of the recordings for the Vgat and Vglut2 Ca^{2+} dynamics to increase the clarity of our data. We now include 60s of the recording instead of 30s as in the original figure. As you can see from the revised figure, the decrease in the Vgat Ca^{2+} activity is clearly visible compared to baseline. We have also plotted area under the curve for this time duration, and this also shows a significant decrease in the area under the curve between BL vs scratch. We have now included this updated analysis in figure 4 of the revised manuscript. Figure below.

3) Discussion: authors should address more of the potential interaction between pain and itch.

Our response: We thank reviewer for her/his suggestion. We have now included an extended discussion addressing the potential interaction between pain and itch in the revised manuscript. We have also added a new summary figure highlighting these potential interactions (see below, this is now Figure 7 in the revised manuscript). These changes can be found in page 14 line 25 to page 15 line 6, and are included below:

“Global chemogenetic inhibition or activation of vIPAG neurons has the same effect on both pain³⁶ and itch behaviors. Namely, global inhibition of the vIPAG potentiates both pain and itch while

global activation of vIPAG neurons attenuates both pain and itch. These results are consistent with the hypothesis that the overall function of PAG output is to inhibit both pain and itch transmission. However, the results of experiments in which we selectively modulated activity of either GABAergic or glutamatergic neurons suggest that the mechanisms by which the PAG modulates pain and itch transmission are more complex, as presented schematically in Figure 7. Pain and itch are known to be processed distinctively at the spinal level. Using cell type specific

chemogenetic manipulations, we found that activating GABAergic or inhibiting glutamatergic neurons in vIPAG causes suppression of itch and potentiation of pain³⁶, whereas inhibiting GABAergic or activating glutamatergic neurons causes potentiation of itch and suppression of pain behaviors. Our findings demonstrate that also at the midbrain level, vIPAG GABAergic neurons and glutamatergic neurons process both pain and itch signals inversely. Thus, although pain and itch are processed and transmitted by similar neuroanatomical substrates at peripheral, spinal and supraspinal sites, they are discriminated and processed distinctly at the cellular level.”

The manuscript would also benefit from more integration of the photometry into the discussion. eg: does scratch relieves itch so it turns down GABA, increases Glu activity? If so, are these cells in a heightened state of activity once treated with chloroquine? A minor thing in most other cases but because it was so prevalent here, it should be addressed more thoroughly.

Our response: We thank the reviewer for this suggestion. We have now updated the discussion section of the revised manuscript to address these issues. These changes can be found on page 13 line 7 to line 20, and are included below:

“We found that vIPAG Vgat+ and Vglut2+ neurons showed opposing responses during chloroquine-evoked scratching in our photometry recordings. Chloroquine evoked scratching inhibits GABAergic neurons and activates glutamatergic neurons. Interestingly, in GABAergic neurons basal activity is elevated just preceding chloroquine evoked scratching, while this phenomenon is not seen during spontaneous scratching bouts. What would be the underlying cause for this? We propose that GABAergic neuronal activity is encoding the sensory and motivational aspects of itch and this activity is suppressed when scratched, possibly contributing to relief of itch. An assumption in this model is that relief of itch by scratching is responsible for suppression of activity of GABAergic neurons. Interestingly, we observed rapid elevated activity in the Vglut2+ neurons subsequent to scratching. These differences in the modulatory effects of GABAergic and glutamatergic neurons on itch response suggests the presence of distinct microcircuit features. It is possible that inhibition of Vgat+ neurons might disinhibit the Vglut2+ve neurons subsequent to scratching during chloroquine-evoked itch. The Ca²⁺ dynamics of Vgat and Vglut2 neurons are consistent with a key role of these neurons in regulating itch processing.”

Minor Things: -Replot figure 1 to match other graphs / show individual data points in all scratch measures in figures 2-5.

Our response: We have now updated all the figures to match figure 1 as suggested by the reviewer.

REVIEWERS' COMMENTS:

Reviewer #1 (Remarks to the Author):

The authors have nicely addressed all of the reviewer's concerns. The revised manuscript is suitable for publication.

Xinzhong Dong

Reviewer #2 (Remarks to the Author):

Authors have performed additional experiments to address my major concern. I think that new data of spinal Fos staining are good addition to this work. The new behavioral test of aversion test is very interesting too. In sum, I think that this work is very important and will have great impact for our understanding of the modulation of itch. I support its publication in the present form.

Reviewer #3 (Remarks to the Author):

The authors were very responsive to my comments.